# Insights into SusCD-mediated glycan import by a prominent gut symbiont

Declan A. Gray[1], Joshua B. R. White[2], Abraham O. Oluwole [3], Parthasarathi Rath[4], Amy J. Glenwright [1], Adam Mazur[4], Michael Zahn[4], Arnaud Baslé[1], Carl Morland[1], Sasha L. Evans[2], Alan Cartmell [5], Carol V. Robinson [3], Sebastian Hiller [4], Neil A. Ranson [2], David N. Bolam [1✉] & Bert van den Berg [1✉]

In Bacteroidetes, one of the dominant phyla of the mammalian gut, active uptake of large nutrients across the outer membrane is mediated by SusCD protein complexes via a "pedal bin" transport mechanism. However, many features of SusCD function in glycan uptake remain unclear, including ligand binding, the role of the SusD lid and the size limit for substrate transport. Here we characterise the β2,6 fructo-oligosaccharide (FOS) importing SusCD from *Bacteroides thetaiotaomicron* (Bt1762-Bt1763) to shed light on SusCD function. Co-crystal structures reveal residues involved in glycan recognition and suggest that the large binding cavity can accommodate several substrate molecules, each up to ~2.5 kDa in size, a finding supported by native mass spectrometry and isothermal titration calorimetry. Mutational studies in vivo provide functional insights into the key structural features of the SusCD apparatus and cryo-EM of the intact dimeric SusCD complex reveals several distinct states of the transporter, directly visualising the dynamics of the pedal bin transport mechanism.

[1] Biosciences Institute, The Medical School, Newcastle University, Newcastle upon Tyne NE2 4HH, UK. [2] Astbury Centre for Structural Molecular Biology, Faculty of Biological Sciences, University of Leeds, Leeds LS2 9JT, UK. [3] Physical and Theoretical Chemistry Laboratory, University of Oxford, Oxford OX1 3QZ, UK. [4] Biozentrum, University of Basel, Basel, Switzerland. [5] Institute of Systems, Molecular and Integrative Biology, University of Liverpool, Liverpool L69 7ZB, UK. ✉email: david.bolam@ncl.ac.uk; bert.van-den-berg@ncl.ac.uk

The human large intestine is home to a complex microbial community, known as the gut microbiota, which plays a key role in host biology[1–3]. One such role is to mediate the breakdown of complex glycans, which would otherwise be unavailable to the host because the human digestive tract lacks the necessary enzymes[4–7]. The products of this anaerobic glycan metabolism are short-chain fatty acids, which provide a range of both localised and systemic health benefits to the host[8,9]. In the colon, the microbiota is dominated by two bacterial phyla, the Gram-negative *Bacteroidetes* and Gram-positive *Firmicutes*[1,10]. The *Bacteroidetes* are glycan generalists and individual species often have the capacity to utilise a wide diversity of polysaccharides from plant, microbial and host sources[7,11,12]. The glycan-degrading apparatus in *Bacteroidetes* is encoded by co-regulated gene clusters known as polysaccharide utilisation loci (PULs), with each PUL encoding all of the proteins required for the acquisition and degradation of a specific glycan[7,13]. For example, the model gut symbiont *Bacteroides thetaiotaomicron* (*B. theta*) has 88 predicted PULs[14], although only a limited number have been characterised[12]. Most glycan degradation occurs intracellularly, and import of the substrate molecules across the outer membrane (OM) is mediated by a class of PUL-encoded TonB-dependent transporters (TBDTs) known as SusCs (we propose to re-purpose the term "Sus" for *s*accharide *u*ptake *s*ystem rather than *s*tarch *u*tilisation *s*ystem)[7,12,13,15]. SusC proteins are unique among TBDTs in that they are tightly associated with a SusD substrate-binding lipoprotein[15–17] (Fig. 1a). Recently, we showed that SusCD complexes mediate substrate uptake via a "pedal bin" mechanism[15,17]. The SusC transporter forms the barrel of the bin, while SusD sits on top of the barrel, opening and closing like a lid to facilitate substrate binding. Previous structures of loaded SusCD complexes revealed a bound ligand, which was completely encapsulated by the closed pedal bin, indicating that SusCD transporters may have a size limit for transport[15,17]. An investigation on the prototypical SusCD system revealed that a mutant strain lacking the surface endo-amylase preferentially utilised malto-oligosaccharides with a degree of polymerisation (DP) of 5–16 (DP5–16)[18], suggesting that this is the preferred size range imported by the Bt3701-02 SusCD transporter. Direct evidence for this notion is lacking, however, and this issue has not been explored for any other Sus systems[12]. Furthermore, many other key features of SusCD function remain unclear, including the role of the SusD lid and other conserved structural elements, as well as the identity of the glycan recognition elements of both SusC and SusD.

One example of a typical PUL is the *B. theta* levan utilisation locus, spanning Bt1754-Bt1765[19]. Levan is a fructan polysaccharide that comprises β2,6-linked fructose units, with occasional β2,1 fructose branches and is produced mainly by bacteria as an exo-polysaccharide, but also by some cereal plants such as wheat[20–23]. For the levan utilisation system, the cell surface components are Bt1760 (a GH32 endo-levanase), Bt1761 (a surface glycan-binding protein; SGBP), Bt1762 (SusD) and Bt1763 (SusC)[15,19,24] (Fig. 1a). The cell surface levanase Bt1760 and glycan-binding proteins (Bt1761 SGBP and Bt1762 SusD) have been shown to be specific for levan, with no activity against, or binding to, β2,1-linked inulin-type fructans that are common to many plants[19,25].

In this study, we characterise binding and uptake of β2,6 fructo-oligosaccharides (FOS) by the Bt1762-63 SusCD transporter. Co-crystal structures of the closed complex with bound glycans reveal the residues involved in substrate recognition and suggest that the large binding cavity can contain several substrate molecules, each up to ~2.5 kDa in size, a finding supported by native mass spectrometry (MS) and isothermal titration calorimetry (ITC). Mutational studies in vivo provide insights into the key structural features of the SusCD apparatus and cryo-electron

microscopy (EM) of the intact SusC$_2$D$_2$ complex reveals several distinct states of the transporter, that is, open-open (OO), open-closed (OC) and closed-closed (CC). These structures directly visualise the dynamics of the pedal bin mechanism and suggest that the individual SusCD complexes function independently of each other. Taken together, these results provide important insights into the mechanism of glycan import by SusCD complexes of dominant gut bacteria.

## Results

**A FOS co-crystal structure reveals SusCD residues involved in ligand binding.** In the *B. theta* levan PUL, the periplasmic enzymes are GH32 exo-acting fructosidases that release fructose from the imported β2,6 FOS (Fig. 1a, b)[19,24]. The fructose is recognised by the periplasmic domain of the PUL sensor–regulator BT1754, which upregulates expression of the PUL[19]. Thus, by growing *B. theta* on fructose, large amounts of the Bt1762-63 transporter can be obtained for structural work[15].

A previous crystal structure of Bt1762-63 obtained in the absence of levan substrate revealed a dimeric (SusC$_2$D$_2$) closed state in which the SusC TBDT (Bt1763) lacked the plug domain as a result of proteolytic cleavage[15]. To provide further insight into glycan recognition and transport by SusCD complexes, we determined a co-crystal structure with β2,6-linked FOS using data to 2.69 Å resolution (Fig. 1c–e; Supplementary Table 1). The FOS were generated by partial digestion of levan by Bt1760 endo-levanase, followed by size-exclusion chromatography (SEC) and analysis by thin-layer chromatography (TLC) and MS. In this structure, containing relatively short FOS (~DP6–12), the plug domain is present. Like the oligopeptide ligands in the Bt2261-64 and RagAB structures[15,17], the FOS is bound at the top of a large, solvent-excluded cavity formed by the Bt1762-63 complex (Supplementary Fig. 1). Density for six β2,6-linked fructose units can be assigned in the structure and this was designated as the primary binding site (Fig. 2a). The bound oligosaccharide is compact and has a twisted, somewhat helical conformation, similar to that observed for levantetraose in complex with the endo-levanase[26]. The ligand makes numerous polar contacts with side chains of residues in both Bt1762 and Bt1763 (Fig. 2b). For Bt1762 (SusD), these residues are D41, N43, D67, R368 and Y395, and for Bt1763 (SusC), T380, D383, D406 and N901. In addition, prominent stacking interactions are present between the ring of fructose 2 (Frc 2) and W85 of Bt1762. Interestingly, a β2,1 decoration is present in the bound ligand at Frc 4, and the branch point interacts with the extensive non-polar surface provided by the vicinal disulfide between Cys298 and Cys299 of Bt1762 (Fig. 2b), suggesting that the transporter may have some specificity for FOS with a β2,1 decoration, or alternatively, that the *Erwinia* levan used contains many β2,1 branches such that most of the levanase products are decorated. Previous studies have shown the proportion of β2,1 linkages in levan from *Erwinia herbicola*[23] to be ~10%, suggesting that ~2 out of every 3 β2,6 hexasaccharides would be decorated.

Surprisingly, the structure contains a second ligand molecule at the bottom of the SusC cavity, contacting the plug (FOS2; Fig. 1d, e, left panel). Density for four fructose units can be seen in this secondary binding site, and polar contacts are made with Bt1763 residues Q372, K454, E481 and W483 in the barrel wall, and with H169 and E170 in the plug domain (Fig. 2c). The fit to the FOS2 density is slightly better for a 3-mer with a β2,1 decoration compared to a β2,6-linked 4-mer. The relatively small size of the co-crystallised FOS, combined with the relative orientation and the large distance between FOS1 and FOS2 (>20 Å; Fig. 1c), makes it highly plausible that there are two ligand molecules in the Bt1762-63 cavity.

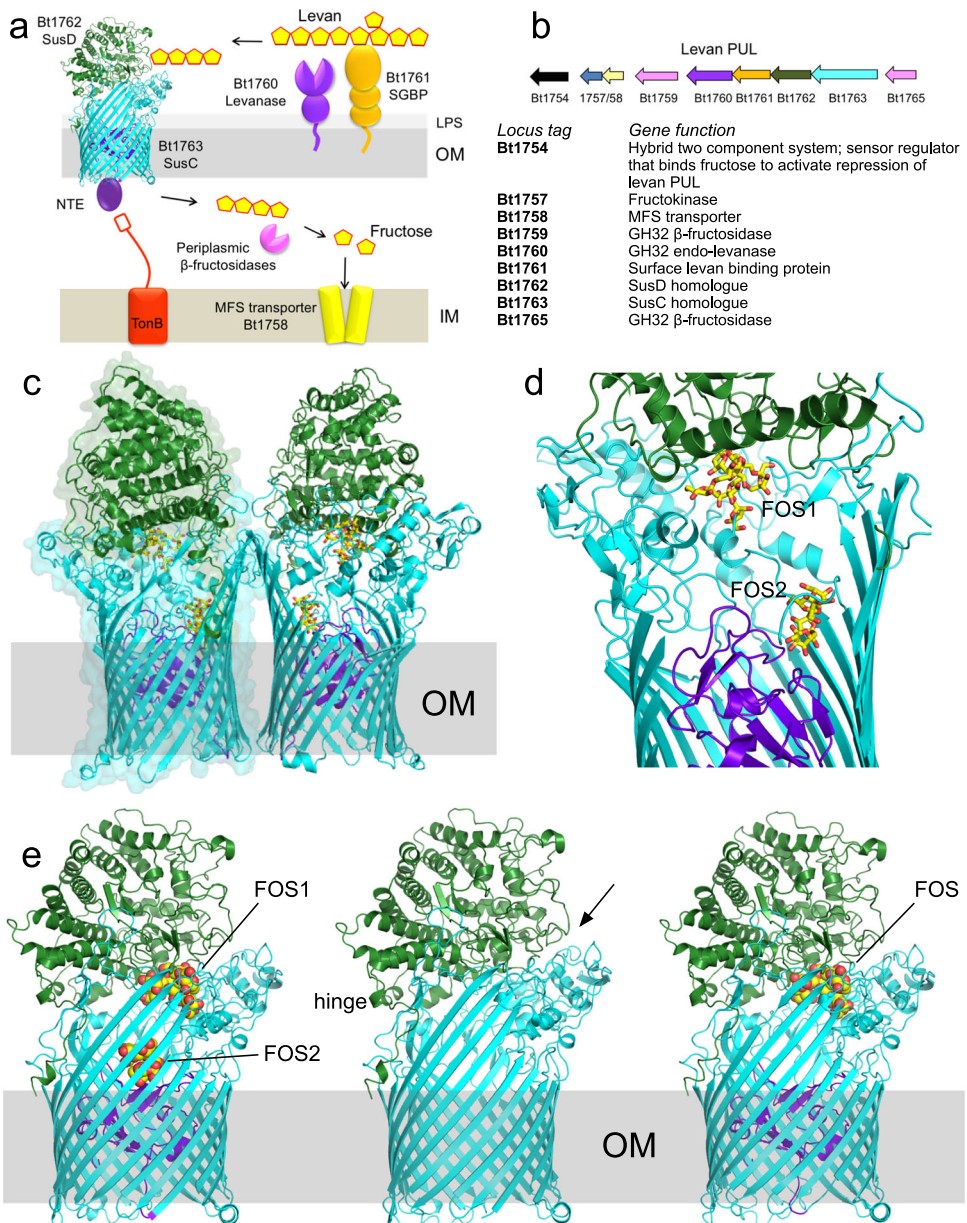

**Fig. 1 Structures of Bt1762-63 in the presence of β2,6 FOS. a, b** Schematic overview of the *B. theta* levan Sus apparatus (**a**), encoded by the PUL spanning Bt1754-65 (**b**). NTE is the N-terminal extension of Bt1763. Bt1759 and Bt1765 are periplasmic GH32 β-fructosidases, Bt1757 is a predicted fructokinase and Bt1758 is a predicted inner membrane (IM) major facilitator superfamily (MFS) transporter. Bt1754 is a hybrid two-component system that binds directly to fructose in the periplasm to upregulate expression of the PUL (**b**)[19]. **c** Cartoon representation of the Bt1762-63 dimer with bound FOS molecules shown as stick models, with oxygens coloured red and carbons yellow. Bt1762 (SusD) is coloured green and Bt1763 (SusC) cyan. The plug domain of Bt1763 is coloured purple. **d** Close-up cut-away view of the two FOS molecules inside one of the BT1762-63 monomers. **e** Cartoon comparisons of Bt1762-63 with shorter FOS (left; PDB ID 9ZAZ; 2.69 Å resolution), without ligand (middle panel; PDB ID 6Z8I; 2.62 Å resolution), and with longer FOS (right; PDB ID 6Z9A; 3.1 Å resolution). The bound FOS molecules are shown as space-filling models. The N-terminal hinge at the back of BT1762, allowing opening of the transporter at the front (arrow), is labelled for the complex without ligand. Crystal structure figures were made with Pymol (https://pymol.org/2/).

Next, we obtained a structure without a substrate using a protein preparation that did not suffer from proteolysis (Fig. 1e, middle panel; Supplementary Table 1). Interestingly, while this structure is very similar to that reported earlier[15], density for the plug domain is poor but clearly present (Supplementary Fig. 2), suggesting mobility or different conformations of the plug within the barrel. While there is no density supporting this notion, it is also possible that the plug has been ejected from some of the Bt1763 barrels in the crystal. Consistent with the relatively high solvent content of the crystals (~63%), there are large spaces underneath the barrels that could contain ejected and mobile plug

domains (Supplementary Fig. 2d). This would allow inadvertent proteolytic cleavage of the plug domain as observed in our previous apo-Bt1762-63 structure (PDB ID 5T3R)[15].

We next added longer β2,6 FOS (~DP15–25) to the same protein preparation used for the apo structure determination, and obtained another co-crystal structure of Bt1762-63, using data to 3.1 Å resolution (Supplementary Table 1). This structure has very similar density for a FOS molecule at the principal binding site (Fig. 1e, right panel) compared to the higher resolution structure with shorter FOS, suggesting that the additional fructose units in longer FOS contribute little if anything to binding. The co-crystal

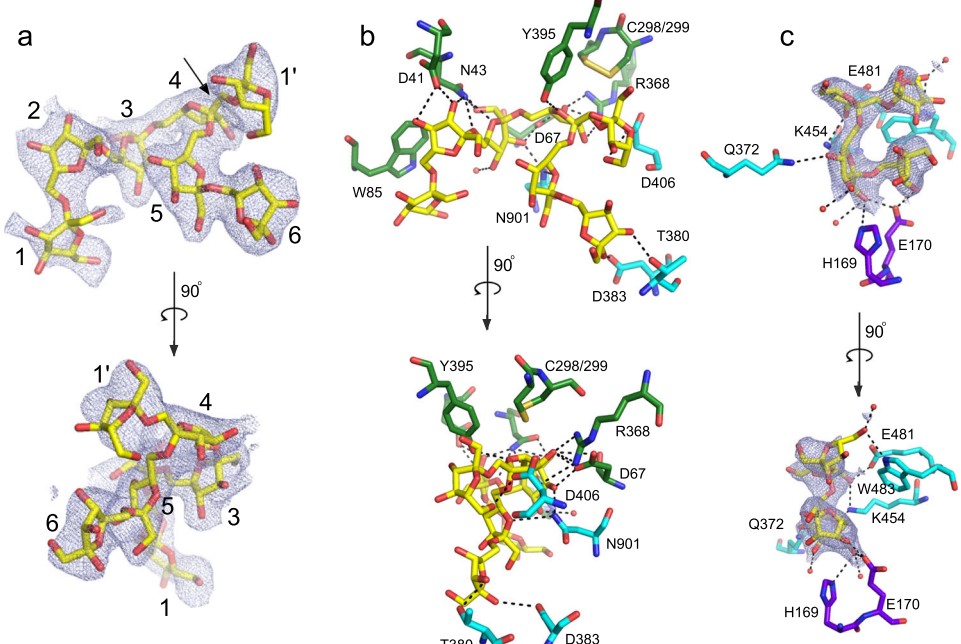

**Fig. 2 FOS densities and interactions with the transporter. a** 2Fo-Fc density (contoured at 1.0$\sigma$, carve = 2) of FOS1 bound in the primary binding site rotated by 90°. The glycan branch point is indicated with an arrow. **b** Close-ups of the principal FOS binding site with residues making polar interactions with the substrate shown as stick models and coloured as in panel (**b**). Polar interactions are shown as dashed lines, and water molecules as red spheres. For both Bt1762 and Bt1763, residue numbering is for the mature protein without the signal sequence, that is, starting at C19 for Bt1762 and G26 for Bt1763. **c** Close-ups of the FOS2 binding site with polar interactions indicated. Plug residues are coloured purple.

structure with the longer FOS also shows some density at the secondary site, but it is of insufficient quality to allow model building, perhaps due to the lower resolution.

**Single-particle cryo-EM reveals different conformational states of Bt1762-63.** The very similar crystal structures of Bt1762-63 in the absence and presence of FOS appear at odds with the dynamics proposed in the pedal bin transport model[15], but are likely explained by the fact that crystallisation selects for compact, stable states. To visualise potential different conformational states, we used single-particle cryo-EM on detergent-solubilised apo-Bt1762-63. Following initial two-dimensional classification, it is clear that Bt1762-63 is dimeric in solution, demonstrating that the dimers in the crystal structures are not artefacts. Strikingly, several distinct conformations of the SusC$_2$D$_2$ complex are present in a single dataset, and after further classification steps, three dominant states were identified. Following three-dimensional (3D) reconstruction, electron density maps were obtained at resolutions allowing rigid body placement and refinement of individual protomers. This yielded structures corresponding to the three possible combinations of open and closed dimeric transporters: OO (3.9 Å), OC (4.7 Å) and CC (4.2 Å) (Fig. 3, Supplementary Fig. 3a and Supplementary Table 2).

With the exception of a few Bt1763 loops (L6 and L7; Supplementary Fig. 3b) and the Bt1762 segment following the lipid anchor that serves as a pivot around which Bt1762 moves upon lid opening, the conformational changes between the three states correspond to rigid body movements of Bt1762. In contrast to the recently characterised RagAB peptide transporter from *Porphyromonas gingivalis*[17], the Bt1762-63 cryo-EM dataset shows evidence of intermediate states of lid opening (Supplementary Fig. 4), suggesting the presence of a number of minima in the energy landscape for opening and closing of the transporter. Interestingly, the presence of the plug domain in the interior of the Bt1763 barrel in solution is strictly correlated

with the open vs. closed state of the transporter; only open states contain a plug domain, regardless of the state of the SusC$_2$D$_2$ dimer (Fig. 3b). Given that there are no direct contacts between Bt1762 and the Bt1763 plug domain, this suggests that the transition from the open to the closed state generates conformational changes in the barrel of Bt1763, leading to displacement of the plug domain from the barrel in vitro. Superposition of the crystal structures of plug-less and complete transporters indeed show non-uniform displacements of up to ~2–2.5 Å for C$_\alpha$ atoms of barrel wall residues, which is substantially more than the coordinate errors of the structures (~0.5 Å), showing that the barrels have subtly different shapes (Supplementary Fig. 5).

The established model of TonB-dependent transport[27] assumes that extracellular substrate binding to a site that includes residues from the plug domain induces a conformational change that is transmitted through the plug domain of the TBDT. This results in disordering of its N-terminal Ton box and increased accessibility of the Ton box for interaction with the C-terminal domain (CTD) of TonB. In the open, apo state of the SusCD transporter as observed via cryo-EM, the Ton box of Bt1763 ([82]DEVVVTG[88]) is visible, tucked away inside the barrel, and interacts with the body of the plug (Supplementary Fig. 3c). By contrast, the density in the crystal structure of the FOS-bound Bt1763 starts only at L97, indicating that the Ton box is disordered and would likely be exposed to the periplasmic space. Thus, our structures for Bt1762-63 are consistent with the proposed behaviour of the Ton box upon ligand binding by the transporter.

**Structure–function studies of Bt1762-63**

*SusD mutants.* The observed interactions of FOS1 with Bt1762 SusD in the co-crystal structures are in excellent agreement with previous ITC studies of levan binding to recombinant Bt1762 variants with single substitutions of aromatic residues and cysteines (Fig. 2b)[15]. The mutations W85A and C298A abolished in vitro levan binding by Bt1762, whereas the affinity of Y395A

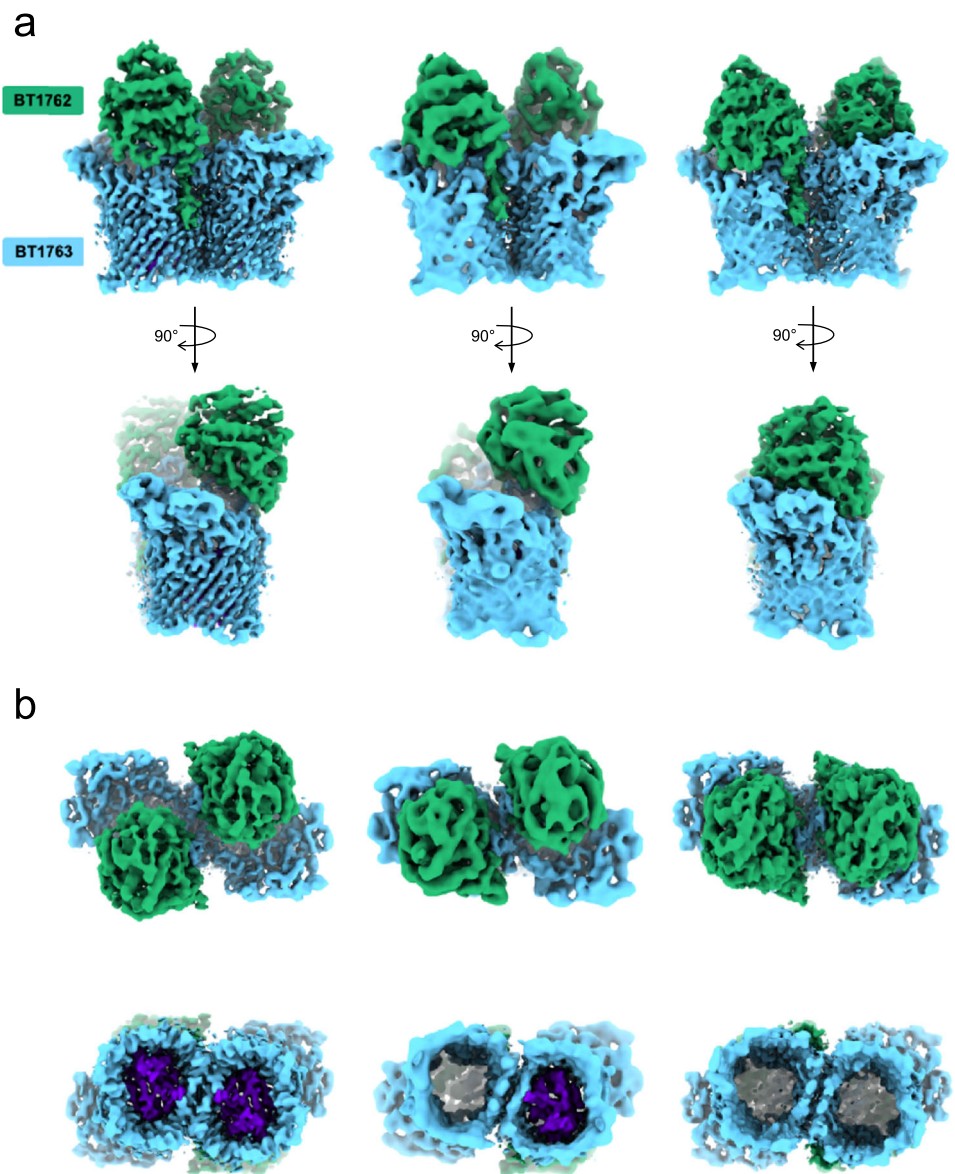

**Fig. 3 Dynamics of pedal bin opening revealed by single-particle cryo-EM. a** Electron density filtered by local resolution for Bt1762-63 viewed from the OM plane for the OO state (left panels), OC state (middle panels) and CC state (right panels). **b** Presence or absence of the Bt1763 SusC plug domain in the three principle states from cryo-EM. Local resolution filtered EM density for Bt1762-63 in the open-open (left), closed-open (middle) and closed-closed (right) states as viewed from the extracellular (top panels) and periplasmic side of the outer membrane (bottom panels). Clear density for the plug domain (purple) is invariably associated with the open state of the transporter. Where the lid is closed, no density is observed within the barrel of Bt1763. Cryo-EM figures were made with ChimeraX[65].

was ~6-fold weaker than that of wild-type (WT). Single substitutions of five other aromatic and cysteine residues remote from the ligand binding site did not affect levan binding[15]. To assess how single substitutions that abolish levan binding to Bt1762 in vitro affect the ability of *B. theta* to utilise levan in vivo, a W85A mutant strain was constructed and growth was analysed using 0.5% *Erwinia* levan as the sole carbon source. Surprisingly, there was no growth defect observed for the W85A strain (Fig. 4a), which, combined with its WT levels of expression (Fig. 4d, e), indicates this variant is fully functional and that care is required in drawing conclusions from in vitro binding studies, especially of the SusD protein in isolation. Clearly, the context of the intact transporter ensures that the effects of SusD point mutations may be much less dramatic in vivo. When seven residues in Bt1762 involved in direct interaction with FOS1 (Fig. 2b; green sticks; D41, N43, D67, W85, C298, R368 and

Y395) were changed to alanine at the same time, the resulting strain (SusD$_{BR}$) grew only after a prolonged lag phase of ~8 h, which is comparable to a strain lacking Bt1762 (ΔSusD; Fig. 4a). However, the OM levels of the SusD$_{BR}$ mutant are much lower than that for WT and W85A strains (Fig. 4d, e), providing an explanation for the observed growth defects. Given these results for Bt1762 mutants, we opted not to investigate the effect of Bt1763 binding residue mutations, but instead focused on other potentially important functional regions of the transporter.

*Hinge region mutants.* Loop L7 of Bt1763, identified as a "hinge" loop in previous simulations of Bt2263-64[15] and hereafter named hinge1, is likely to be important for Bt1762 lid opening (Supplementary Fig. 3b). In addition, we identified from the cryo-EM structures a second loop in Bt1763, L6 (Supplementary Fig. 3b), as potentially important for lid opening (designated as hinge2).

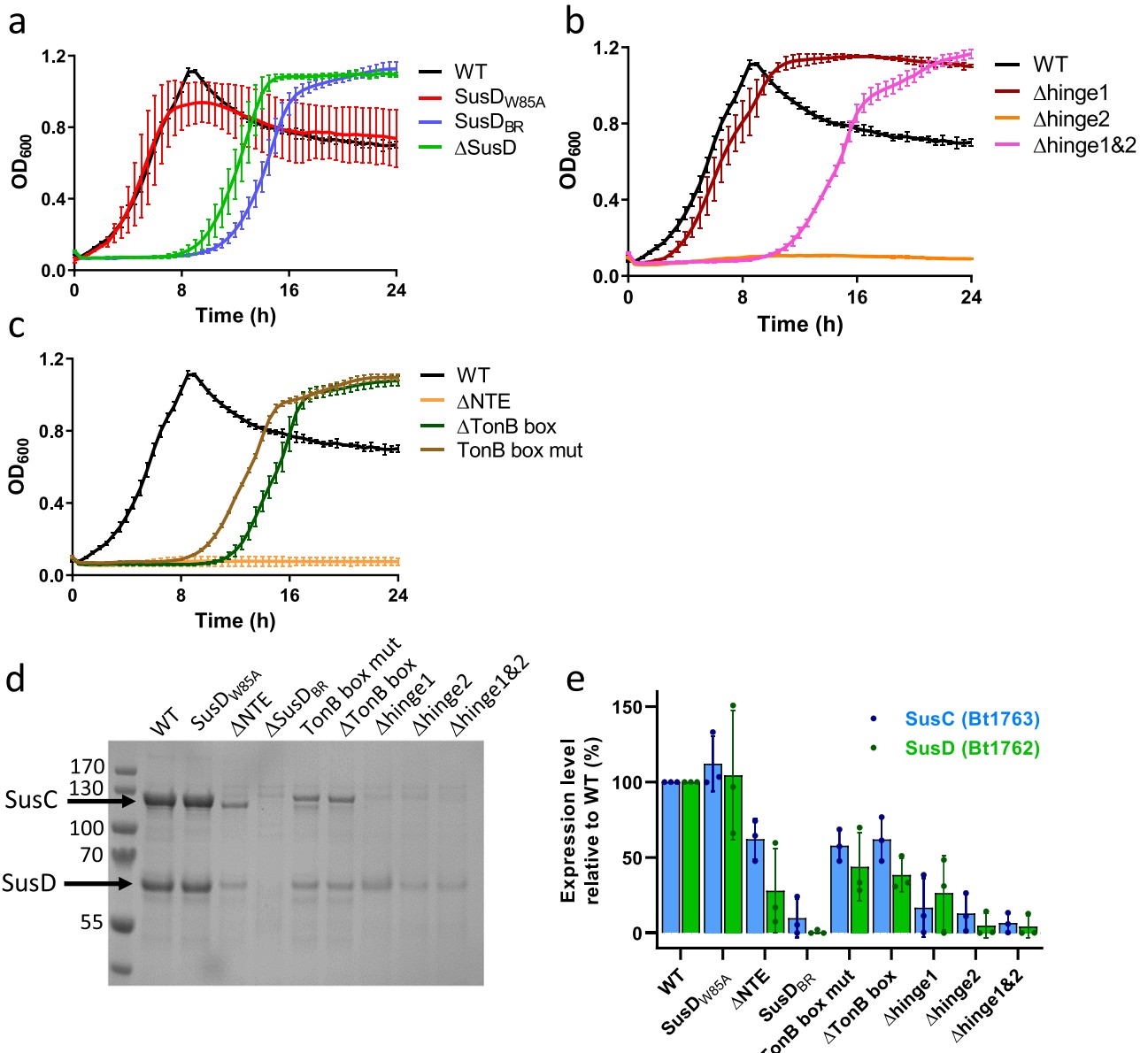

**Fig. 4 Structure–function studies of the Bt1762-63 SusCD levan importer. a–c** Growth curves on levan of Bt1762 SusD mutant strains (**a**), Bt1763 SusC hinge loop mutant strains (**b**), and Bt1763 ΔNTE and TonB box mutant strains (**c**). WT wild-type strain (with His6 tag on C terminus of Bt1762). Growth curves show mean (solid line) ± SD of triplicate wells and are representative of three independent experiments. **d** Representative SDS-PAGE gel (*n* = 3 independent experiments) of IMAC-purified Bt1762-63 complexes from OM fractions of WT (Bt1762-his strain) and mutant strains grown on MM fructose. The far left lane is MW marker with the sizes shown in kDa. **e** Relative expression levels quantified from band intensities on SDS-PAGE gels using ImageJ (*n* = 3 independent experiments; means ± SD are shown, plus individual values). Band intensities of WT Bt1762 and Bt1763 were set to 100%. Source data are provided as a Source Data file.

Hinge1 and 2 change conformation substantially during lid opening and are responsible for the majority of interactions between Bt1762 and Bt1763 in the open state (Supplementary Fig. 3b). The deletion of hinge1 caused little to no growth defect (Fig. 4b), which is surprising given that Bt1762-63$_{\Delta hinge1}$ expression is very low (Fig. 4d, e) and suggesting that this mutant might have a higher transport activity than the WT. By contrast, the Δhinge2 strain showed a complete lack of growth during the 24 h monitoring period, while being expressed at levels similar to Bt1762-63$_{\Delta hinge1}$ or slightly lower (Fig. 4b, d, e). Surprisingly, a strain in which both hinges were deleted (Bt1762-63$_{\Delta hinge1\&2}$) grew similarly to the ΔSusD strain, that is, after an ~8 h lag phase (Fig. 3b). There is no clear explanation for these phenotypes, since the expression levels of the double hinge mutant are similar

to those of the single hinge deletions. One possibility is that the loss of both Bt1763 hinges allows free movement of the Bt1762 lid, which will only be attached to Bt1763 via the N-terminal ~15 residue segment following the Bt1762 lipid anchor. This may essentially decouple substrate binding by Bt1762 from delivery to Bt1763, and result in a phenotype that is similar to that of ΔSusD. We speculate that the complete loss of growth observed in the Bt1762-63$_{\Delta hinge2}$ mutant could be due to a "dominant-negative" effect on something that does not involve the levan PUL but is essential for growth, perhaps arising from misfolding of Bt1763 that leads to stalling of the BamA assembly machine.

*TonB box and N-terminal extension mutants.* SusC-like proteins are predicted to be TBDTs, but direct evidence for this is lacking.

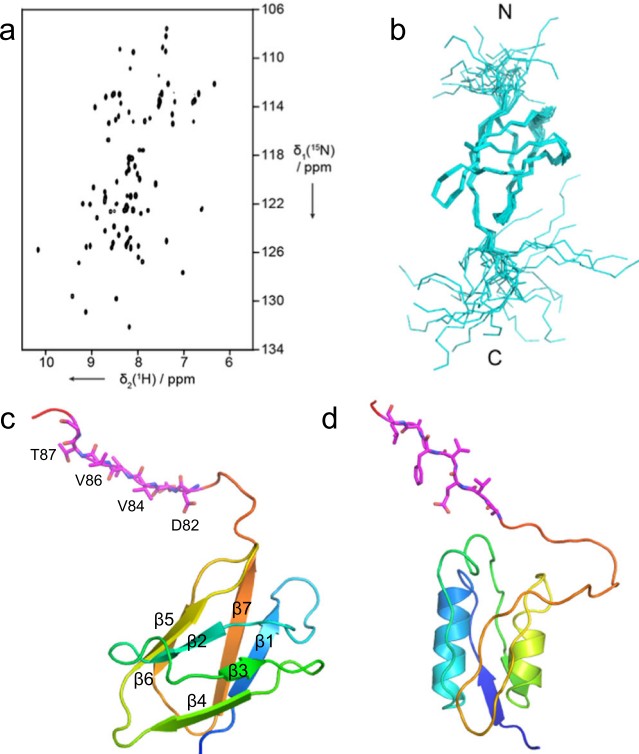

**Fig. 5 Solution NMR structure of the Bt1763 NTE shows a 7-stranded barrel. a** 2D [$^{15}$N,$^{1}$H]-HSQC spectrum of the NTE. **b** Ensemble of the 20 lowest-energy structures of the NTE presented in stick view for the main chain. The N and C termini are labelled. **c** Lowest-energy NMR structure from the ensemble of panel (**b**) in ribbon representation, coloured in rainbow mode (N terminus; blue). **d** STN domain of *Pseudomonas aeruginosa* FoxA (PDB ID 6I97) in the same colouring mode. The Ton boxes of each domain ($^{82}$DEVVVTG$^{88}$ in Bt1763; based on the alignment with FoxA) are coloured magenta.

We therefore examined the importance of the putative TonB box located at the N terminus of Bt1763. In addition, SusC proteins have an N-terminal extension (NTE) domain of ~9 kDa (Pfam 13715, carboxypeptidase D regulatory-like domain) that is absent in other types of TBDTs and precedes the TonB box, but has an unknown function (Fig. 1a)[28]. We constructed two mutant *B. theta* strains in which the TonB box was either deleted (residues $^{81}$VDEVVV$^{86}$; ΔTonB box) or mutated to all-Ala (TonB box mut), and a strain in which the complete NTE was deleted (residues 3–76; ΔNTE), but with the Ton box intact. Growth of the mutants on levan as the sole carbon source revealed that both TonB box mutant strains had similar growth defects to that of ΔSusD, and were expressed at reasonable levels (~50% of WT; Fig. 4c–e), demonstrating that Bt1763 is a bona fide TBDT. This result also suggests that all mutants with growth lag phases longer than 8 h are non-functional, and that the observed growth in these cases is caused by a build-up of small FOS products generated by the surface endo-levanase, which enter the cell via passive diffusion independently of levan PUL components. Remarkably, the ΔNTE strain has a similar phenotype to Bt1762-63$_{Δhinge2}$ with no detectable growth after 24 h, despite expression levels of ~25–50% that of WT, indicating an important role for this domain in transporter biogenesis or cellular function that perhaps might not involve the transport process itself.

**Solution NMR structure shows an immunoglobulin (Ig)-like fold for the NTE.** The striking, dominant-negative effect of the

ΔNTE mutant on growth made it important to determine the structure of the isolated NTE, since it is invisible in the X-ray structures and has a very weak density in the cryo-EM structures. Crystallisation trials proved unsuccessful, and we, therefore, solved the structure by high-resolution nuclear magnetic resonance (NMR) spectroscopy, using uniformly $^{15}$N,$^{13}$C-labelled protein produced in *Escherichia coli* (Supplementary Fig. 6). A 2D [$^{15}$N,$^{1}$H]-HSQC spectrum of the NTE showed good chemical shift dispersion and the expected number of resonances (Fig. 5a). Complete sequence-specific assignments for the backbone resonances were then obtained using a combination of 2D and 3D experiments (Supplementary Tables 3 and 4 and Supplementary Fig. 6). Backbone chemical shifts were compared to the random coil values, resulting in the identification of seven β-strands. A total of 964 distance constraints were obtained from 3D nuclear Overhauser effect spectroscopy (NOESY) spectra, and served as input for structure calculations in CYANA. The final ensemble of 20 lowest-energy structures had a root-mean-square deviation (r.m.s.d.) of 0.52 Å for the backbone heavy atoms (Fig. 5b, Supplementary Fig. 6 and Supplementary Tables 3 and 4). The NTE structure shows a well-defined core of an Ig-like fold with a 7-stranded barrel (Fig. 5c, left panel). The N terminus (including the His-tag) and the C terminus, corresponding to the Ton box, are flexibly unstructured, as evidenced from their random coil chemical shifts and the absence of long-range NOEs (Fig. 5b and Supplementary Fig. 6c).

A distance-matrix alignment (DALI) analysis[29] returned a number of structures with significant similarity, the highest of which had a Z-score of 10.2 (transthyretin-like domain of carboxypeptidase D, PDB ID 5aq0; Supplementary Fig. 7). However, none of the DALI hits provided any insights into a potential function of the NTE. In Proteobacteria, structures of TBDTs with a different N-terminal domain, designated STN (Pfam 07660), have been reported. This STN domain has a similar size as the NTE and has an established role in signalling, where it interacts directly with an anti-sigma factor in the inner membrane to stimulate TBDT expression in response to the presence of their cognate substrates[27,30–33]. Notably, SusCs from ECF-sigma/anti-sigma controlled PULs also contain STN domains in addition to an NTE domain, with the NTE preceding the STN in all cases[34]. STN domains in SusCs have also been shown to be involved in anti-sigma signalling, indicating that the NTE domain in SusCs has a different role compared to the STN[34]. The structure of the FoxA STN in complex with the CTD of TonB shows that the structure of the STN is different from that of the NTE, with the STN containing two helical segments and having a different topology (Fig. 5c, d)[35]. In both domains, the Ton box is separated from the domain body and will be accessible to binding by the CTD of TonB. One intriguing possibility for a role for the NTE could be to provide interaction specificity for the multiple TonB orthologs present in the *B. theta* genome.

**Investigation of the size range of FOS import by Bt1762-Bt1763.** The co-crystal structures of Bt1762-63 with FOS, as well as the previously published structures of Bt2261-64[15] and RagAB[17], show that ligands are encapsulated within a large, solvent-inaccessible cavity formed by the closure of the SusD lid, strongly suggesting that there is a size limit for substrate import. This hypothesis is further supported by previous studies showing that growth of *B. ovatus* and *B. thetaiotaomicron* on large polysaccharides, such as xylan and starch, requires the action of PUL-encoded surface endo-acting enzymes to cleave the glycans into smaller oligosaccharides prior to import[36,37]. In addition, uptake of monosaccharides and even very small oligosaccharides does not appear to require a functional Sus apparatus, suggesting that

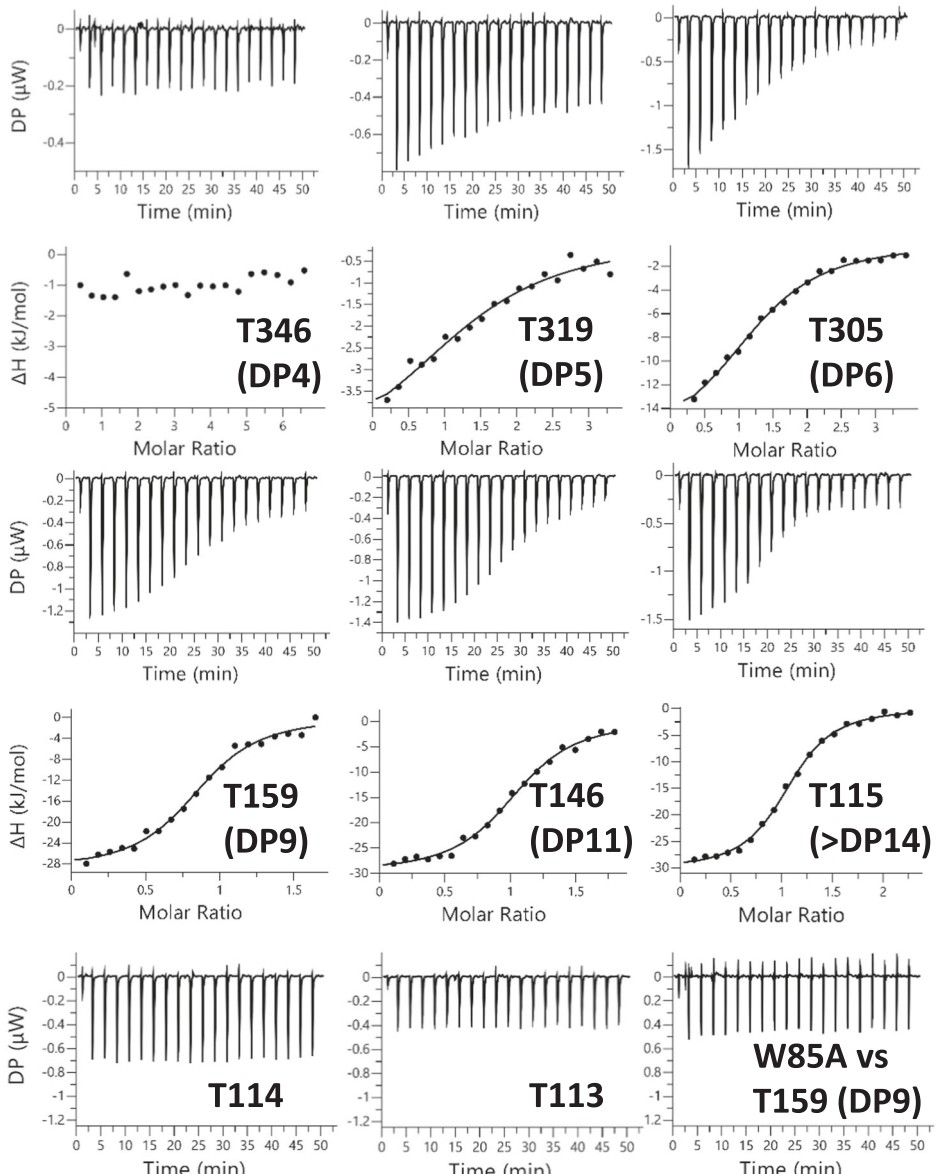

**Fig. 6 FOS binding to Bt1762-63 analysed by ITC.** β2,6 FOS fractions from SEC of partially digested *Erwinia* levan were titrated into purified Bt1762-63 SusCD (25 or 50 µM) in 100 mM HEPES, pH 7.5 containing 0.05% LDAO. The identity of the fraction used is indicated as is the approximate DP of the main FOS species present as determined by MS. The W85A mutant is for SusD in the complex (i.e. Bt1762$_{W85A}$-63). The upper parts of each titration show the raw heats of injection and the lower parts the integrated heats fit to a one set of sites model if binding was observed.

there may also be a minimum size limit for transport by SusCD complexes[38]. To shed light on this issue, we used ITC to investigate the size range of FOS recognised by Bt1762-63. The β2,6 FOS was produced by digestion of *Erwinia* levan with Bt1760 endo-levanase, followed by separation via SEC ("Methods"). For estimation of the DP of each oligosaccharide, we used both TLC and MS (Supplementary Fig. 8). While this was effective for FOS with a DP 4 to ~12, FOS larger than ~DP14 do not move from the origin in TLC, and the MS analysis of FOS larger than ~DP12 proved difficult because of poor ionisation efficiency of longer oligosaccharide molecules. It is however clear that the higher DP FOS fractions contain a wide range of sizes.

ITC revealed that Bt1762-63 binds to the majority of FOS examined, with only the tetra-saccharide displaying no affinity for the transporter (Fig. 6, Supplementary Fig. 9 and Supplementary Table 5). For the larger FOS, affinity increased from DP5–6 ($K_d$ ~ 30 and 17 µM, respectively) and plateaued at DP8 (tube 174, T174

$K_d$ ~ 1 µM), with Bt1762-63 binding to all FOS between DP8 and at least DP13–14 (T115) with similar affinity. These data are in broad agreement with the co-crystal structures, which show well-defined density for 7 fructose units in the primary binding site, suggesting that these provide the bulk of the binding interactions. Surprisingly, no binding was detected for the FOS in SEC fractions T114 and T113, despite these fractions having similar MS profiles to T115 with a broad range of oligosaccharides present (Fig. 6 and Supplementary Fig. 8). The average molecular weight of the FOS in tube T114 ($M_n$ > 2666) is larger than that of T115 ($M_n$ > 2193; Supplementary Fig. 8), and it may be that this increase in average size is enough to preclude binding to the transporter. Furthermore, based on the co-crystal structure, we can see that at least some, and perhaps all, of the bound *Erwinia* levan-derived FOS has a β2,1 decoration, which may influence binding to Bt1762-63. However, it was not possible to identify β2,1 decorations in the TLC or MS analysis. Thus, T115 FOS

might contain significantly more branched species than T114 and this could explain a higher affinity for the T115 fraction. Taken together, however, these data indicate that there is both an upper and lower size limit for FOS binding to the Bt1762-63 transporter in vitro, with the lower limit being DP5 and the upper limit ~DP15. In addition to WT Bt1762-63, we also measured binding of ~DP9 FOS to the Bt1762$_{W85A}$-63 variant (Fig. 6). Surprisingly, no binding is observed for the mutant, even though the Bt1762$_{W85A}$-63 strain grows as well as WT on levan (Fig. 3a), suggesting that a native level of FOS binding by Bt1762-63 is not essential for Bt1762-63 function in vivo. A possible explanation for this observation is that in the presence of high concentrations of FOS, substrate binding to Bt1762 is not needed to load the Bt1763 transporter. The contrasting, loss-of-function phenotype of SusD$_{BR}$ can be explained by the very low expression levels for this mutant (Fig. 4).

We used native MS to shed further light on the ligand binding property of the Bt1762-63 complex. For the ligand-free protein, the mass spectrum revealed charge states corresponding to the intact SusC$_2$D$_2$ dimeric complex and low-intensity peaks assigned to SusCD monomer (Supplementary Fig. 10). Ligand binding appeared to induce considerable destabilization of the SusC$_2$D$_2$ complex as reflected in the higher intensity of monomer peaks in the presence of FOS molecules. Specifically, binding of the FOS fractions characterised by DP > 14 (T115) and DP > 16 (T114) yielded low-intensity adducts to the intact SusC$_2$D$_2$ consistent with substrate binding for both samples. While this appears surprising, it should be noted that higher ratios of FOS:protein were used in MS compared to ITC (~6-fold vs. ~2-fold, respectively), providing a possible explanation for the lack of observed binding in ITC for T114. The complexity of the binding pattern in the spectrum is consistent with polydispersity of the T114 and T115 fractions (Supplementary Fig. 8). More useful insights were obtained with the T159 sample, which consists mainly of FOS with 7–10 fructose units (~DP9) (Fig. 7 and Supplementary Fig. 8). These medium-chain oligosaccharides bind preferentially to the intact SusC$_2$D$_2$ dimer rather than to the SusCD monomer such that no ligand-free dimer was evident in the spectrum, potentially suggesting some kind of cooperativity for ligand binding in the dimer. Interestingly, the relative proportions of protein-bound FOS mirrored their abundance in the T159 sample (Fig. 7b), supporting the similar affinities of FOS with 8–10 fructose units for Bt1762-63 as measured by ITC (Fig. 6 and Supplementary Table 5). At the higher FOS concentrations, binding of more than one FOS molecule per SusCD transporter was observed (Fig. 7b), confirming the observation from our co-crystal structure that more than one ligand molecule can be present in the binding cavity, at least for the relatively small FOS.

Finally, we wanted to confirm the upper FOS size limit in vivo by testing growth of a strain lacking the surface endo-levanase Bt1760 with FOS of different sizes as the sole carbon source. The Δ1760 strain was previously reported to lack the ability to grow on levan[19], which would provide another indication that the Bt1762-63 complex cannot import high molecular weight substrates. Surprisingly, however, the growth rate of the Δ1760 strain on levan from several different sources was similar or only slightly slower than that of the WT strain (Supplementary Fig. 11). Polymerase chain reaction (PCR) of the Δ1760 cells taken from stationary phase of the cultures confirmed the deletion of the *Bt1760* gene from the cells, indicating that the phenotype was not due to contamination with WT strain (Supplementary Fig. 11). These data suggest that all levans tested contained enough low DP FOS to allow growth without needing digestion by the surface endo-levanase. The spent media collected after culturing the Δ1760 strain did not support growth of the mutant strain after re-inoculation. However, it was not possible to

obtain size estimations of the remaining FOS via either HPAEC-PAD or MS, due to the technical limitations of separation and ionisation, respectively, of FOS at this high DP. It was therefore not possible to determine an upper FOS size limit of the Bt1762-63 importer in vivo.

## Discussion

The difficulty in ionising large oligosaccharides in the mass spectrometer[39] complicates the reliable experimental determination of an upper substrate size limit of SusCD-like systems. Even if large oligosaccharides could be detected via MS, the inability to separate high DP FOS (>~DP12) by SEC precludes estimation of detection efficiency, and consequently quantitation. Despite this, some important insights can be obtained via simple structural considerations. The ligand binding cavity of Bt1762-63 has a volume of ~10,000 Å$^3$ as determined via CASTp[40], which is similar to that of the peptide importer RagAB[17]. While no information is available for FOS, data for sucrose[41] allow an estimation of ~500 Å$^3$ per fully hydrated molecule, with five waters per sucrose. For one bound water per sucrose, the molar volume is lowered by ~20%, leading to a value of 400 Å$^3$ per sucrose. Thus, depending on FOS hydration inside the Bt1762-63 cavity and assuming ideal packing, there would be space for 20–25 sucrose molecules, that is, 40–50 sugar monomers. Given that it is unlikely that the twisted structures of levan FOS could pack very efficiently, we estimate that the maximum number of fructose units occupying the Bt1762-63 cavity at any one time could be no more than ~30–35, putting an upper limit on total bound FOS of ~5 kDa. Considering our ITC and native MS data, it is likely that this total mass would comprise several individual molecules, rather than one large molecule. Given that there are unlikely to be large structural differences among SusCD-like systems, we suggest ~5 kDa as a general total size limit for these transporters, which is consistent with recent data for the archetypal Sus[18].

Our structures that show FOS in the principal binding site at the Bt1762-63 interface raise an important question: how is ligand occupancy relayed to the plug domain, and how does this lead to increased accessibility of the Ton box? This key issue is likely unique to SusCD systems, in particular, those SusCs without the long plug loop present in, for example, Bt2264[15] that is able to contact ligand in the principal binding site and relay binding site occupancy directly to the plug domain (Supplementary Fig. 12). In Bt1763, the smallest distance between the visible part of the substrate in the principal binding site and the plug is 15 Å, and so an optimal-sized FOS (in terms of binding affinity) of ~DP8–12 could not contact the plug directly. The presence of a second substrate molecule at the bottom of the binding cavity (Fig. 1d) might be a way to overcome this problem, implying a mechanism in which the binding cavity "bin" is filled first via two or more substrate binding-release cycles to provide plug contacts, that collectively increase the accessibility of the Ton box and binding to the CTD of TonB.

The substrates for the transporter are generated by the combined action of the Bt1760 endo-levanase and the Bt1761 SGBP (Fig. 1a). It is not yet clear if, and how, these two proteins are spatially and temporally connected to the Bt1762-63 transporter. For the archetypal Sus, data suggest that all five OM components of the PUL (SusCDEFG) are arranged in one stable complex[16,42]. Other, more recent studies paint a much more dynamic picture, with the SGBPs (SusE and SusF) and the endo-amylase SusG transiently associating with the SusCD core complex[43,44]. Besides depending on the type of levan[25], the FOS sizes delivered to Bt1762-63 will depend critically on the binding kinetics of the Bt1760 levanase and on the proximity of Bt1761: a close

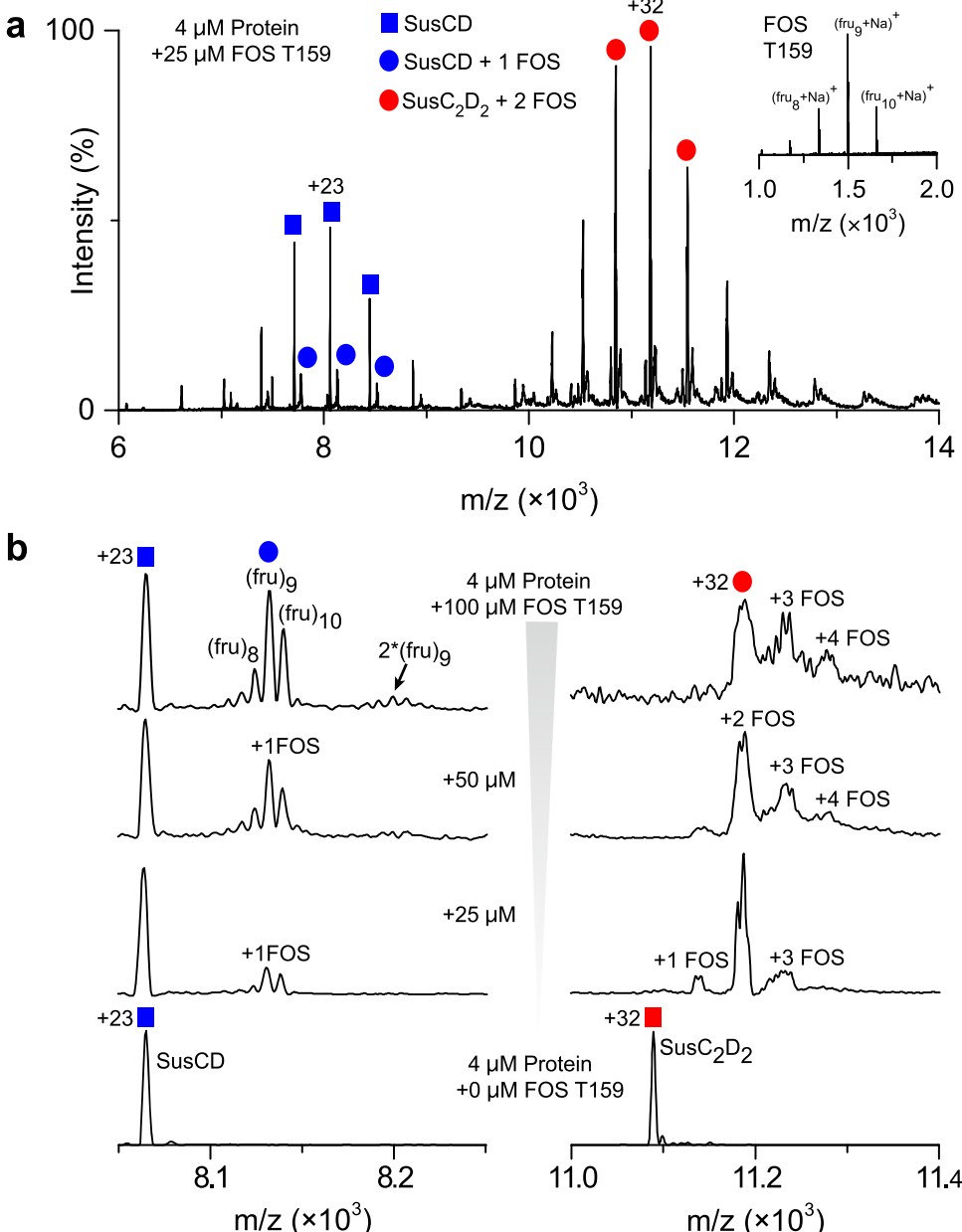

**Fig. 7 Bt1762-63 binding to medium-chain fructo-oligosaccharide (FOS). a** Native mass spectrum of Bt1762-63 bound to FOS T159 fraction (~DP9) revealed the presence of both bound and unbound SusCD monomers, but only FOS-bound SusC$_2$D$_2$ dimers. The inset shows the spectrum of the FOS T159 fraction. **b** Spectrum of Bt1762-63 incubated with different concentrations of FOS T159. Shown are individual charge states of the monomer (+23) and of the dimer (+32). Up to two and four FOS molecules (mainly 8–10 fructose units) were found to bind to SusCD and SusC$_2$D$_2$, respectively. Masses of all species observed are listed in Supplementary Table 6.

association between the two would most likely favour production of uniformly sized FOS of relatively small size, which, as we have shown, are preferred substrates. Likewise, a close association between the enzyme and Bt1762-63 will enhance the capture of the generated FOS by SusD and subsequent delivery to SusC. With regards to this last step, it is interesting to note that, in contrast to in vitro conditions, the substrate binding function of Bt1762 is not necessary in vivo (Fig. 4 and Fig. 6), that is, when all other OM components are present (Bt1760/61/63), and similar data have recently been obtained for other Sus[18,45]. These observations would suggest that the SGBP can assume the substrate-delivery role of SusD and argues in favour of an intimate association of all OM components of a PUL. This then raises the question of why the SusD lid has evolved at all. The fact

that the *presence* of SusD is important in vivo (Fig. 4) suggests that its function as a lid that can open and close is vital.

Our structural data provide important clues about the function of SusD and about glycan import in general (Fig. 8). The basis for these clues is the unprecedented observation that closed, but empty transporters lack the entire plug domain. This spontaneous expulsion of the plug is likely to be non-physiological and caused by a loss of lateral membrane pressure due to detergent solubilisation. Nevertheless, it does suggest that Bt1762 lid closure causes conformational changes within the Bt1763 barrel that decrease the "affinity" of the plug for the barrel. This may facilitate the removal of the entire plug domain from the barrel by TonB action, as opposed to local unfolding and formation of a relatively narrow channel as has been proposed for non-Sus

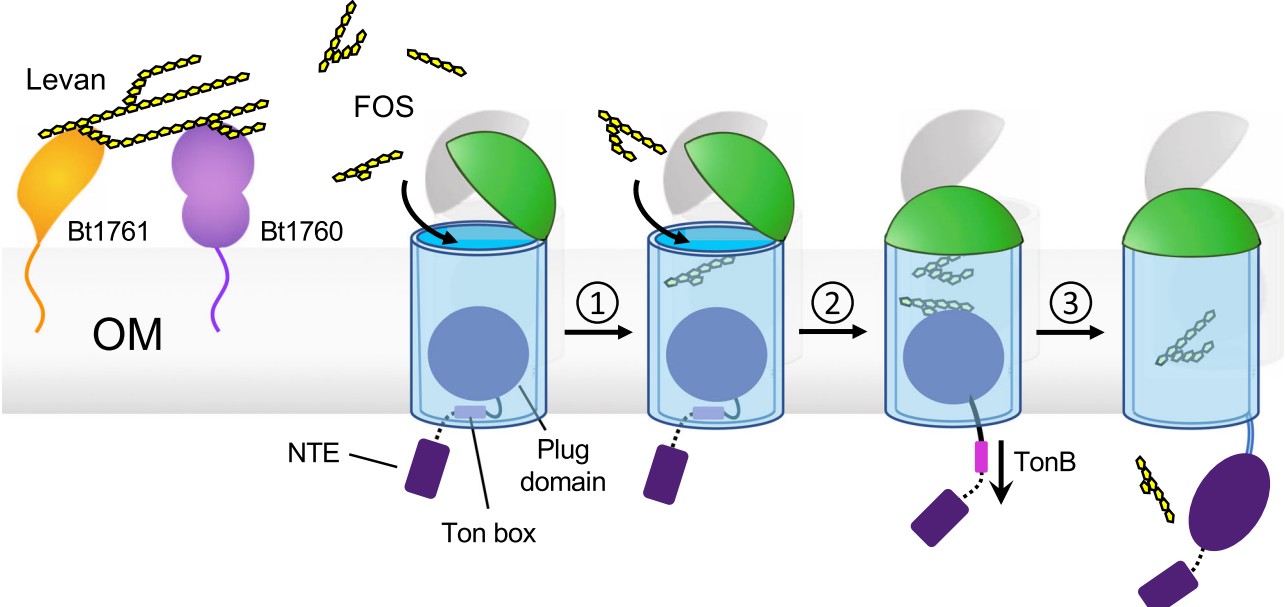

**Fig. 8 Model demonstrating the proposed mechanism of substrate capture and translocation by Bt1762-63.** Levan polysaccharide is initially bound and hydrolysed at the cell surface by the SGBP Bt1761 and GH32 endo-levanase Bt1760, respectively, although the precise role of the SGBP is unclear. A lid-open state of the transporter permits binding of transport-competent FOS. ① Contributions from both SusC and SusD to FOS binding elicits closure of the lid. ② Multiple cycles of lid opening and closing occur until the SusC "bin" is fully loaded with substrate, forming the transport-ready state of the complex. Substrate loading is communicated across the outer membrane by direct contact with the Bt1763 plug domain, inducing perturbation of the Ton box region on the periplasmic side of the plug, rendering it accessible to TonB. ③ TonB-mediated disruption/extraction of the plug permits substrate translocation. Details of the "reset" mechanism are unclear, but reinsertion of the plug is likely a prerequisite to restoring the open state of the transporter.

TBDTs[46–48]. To prevent plug removal in the absence of substrate resulting in futile transport cycles, we postulate that only the direct contact of substrate with the plug (as observed in the co-crystal structure with FOS2 and the in Bt2263/64–peptide complex[15]) leads to increased accessibility of the TonB box and interaction with TonB (Fig. 8). In our model, the impermeability of the OM, which otherwise would be compromised due to the formation of a very large channel of ~20–25 Å diameter, would be preserved by the seal provided by the closed SusD lid. Upon reinsertion of the plug, the transporter would revert back to its open state (Fig. 8). The most important function of SusD proteins during glycan import may therefore be to provide a seal to preserve the OM permeability barrier.

## Methods

**Maintenance and growth of bacterial strains.** Brain heart infusion (BHI) media and agar, supplemented with 1 μg/ml haematin, or where needed with gentamycin (200 μg/ml) and erythromycin (25 μg/ml), was used for routine selection and maintenance of *B. thetaiotaomicron* VPI-5482 (*B. theta*) WT and mutant strains. Luria-Bertani media and agar, supplemented where needed with kanamycin (50 μg/ml) or ampicillin (100 μg/ml), was used for routine selection and maintenance of *E. coli* strains. Cultures for *B. theta* protein purification were grown in a chemically defined minimal media (MM), NH₄SO₄ 1 mg/ml, Na₂CO₃ 1 mg/ml, cysteine 0.5 mg/ml, KPO₄ 100 mM, vitamin K 1 μg/ml, FeSO₄ 4 μg/ml, vitamin B12 5 ng/ml, mineral salts 50 μl/ml (NaCl 0.9 mg/ml, CaCl₂ 26.5 μg/ml, MgCl₂ 20 μg/ml, MnCl₂ 10 μg/ml and CoCl₂ 10 μg/ml) and haematin 1 μg/ml. These cultures were supplemented with D-fructose (0.5% final) as the sole carbon source to yield MM-Frc (MM fructose). Frc activates the levan PUL, but does not require Bt1762-63 for import. All *B. theta* growths were carried out under anaerobic conditions at 37 °C (A35 Workstation, Don Whitley Scientific).

**Gene cloning and construction of *B. theta* mutants.** The desired genetic sequence containing mutations or deletions was amplified through two rounds of PCR, including ~1000 bp upstream and downstream of the deletion or mutation. The primers used to generate these PCR products are listed in Supplementary Table 7. Briefly, for the TonB deletion strain primers TB-fwd, TBD-rev and TBD-fwd, TB-rev respectively were used to make 1000 bp fragments, which underwent an additional PCR to construct the 2000 bp sequence. The same process was used to make mutants TonB mutation (TB-fwd, TBM-rev and TBM-fwd, TB-rev),

Δhinge1 (Hinge-fwd, H1D-rev and HD1-fwd, Hinge-rev), Δhinge2 (Hinge-fwd, HD2-rev and HD2-fwd, hingerev), W85A (Trp1-fwd, Trp2-rev and Trp3-fwd, Trp4-rev), and ΔNTE (NTE1-fwd, NTE2-rev and NTE3-fwd, NTE4-rev).The final PCR product was ligated into pExchange-tdk vector[38].

*B. theta* genetic deletions and mutations were created by allelic exchange using the pExchange-tdk vector. Briefly, the constructed pExchange-tdk plasmids, containing the mutations/deletions plus ~1000 bp flanks up- and downstream, were transformed into S17 λ pir *E. coli* cells, in order to achieve conjugation with the *B. theta* recipient strain[38]. The conjugation plates were scraped to generate a culture containing *B. theta* and *E. coli*. *B. theta* cells undergoing a single recombination event were selected by plating on BHI-haematin agar plates containing gentamycin (200 μg/ml) and erythromycin (25 μg/ml), and 8–12 colonies were restreaked on fresh BHI-haematin-gent-ery plates. Single colonies were cultured in BHI-haematin and pooled. To select for the second recombination event, pooled cultures were plated on BHI-haematin agar plates containing 5-fluorodeoxyuridine (FUdR; 200 μg/ml). FUdR-resistant colonies, 8–12, were restreaked on fresh BHI-haematin-FUdR. From these, single colonies were cultured in BHI and genomic DNA was extracted and screened for the correct mutations using diagnostic PCR and sequencing. DNA-sequencing was performed by MWG-Eurofins.

**Expression and purification of native proteins.** The required strain of *B. theta* was inoculated from −80 °C stocks into 5 ml BHI supplemented with haematin, 1 μg/ml. These were cultured overnight and used to inoculate 500 ml Duran bottles containing 500 ml MM supplemented with haematin and either fructose or levan (0.5%) as the sole carbon source and grown for 18–20 h. The following morning, cells were harvested by centrifugation at 11,305 × g for 25 min, and the pellets were resuspended in Tris salt buffer (TSB; 20 mM Tris, 300 mM NaCl, pH 8) and stored at −20 °C.

One protease inhibitor tablet (Complete Mini, EDTA free; Roche) and DNaseI (Roche) were added to the completely thawed cells from 4 l of culture before (total volume ~100 ml) being lysed in a cell disrupter (Constant Systems; 0.75 kW model) at 23 kilopounds per square inch. The lysed cells were centrifuged for 1 h at 204,526 × g (45Ti rotor, Beckman). The resultant pellet, containing total membranes, was homogenised in 100 ml 0.5% sodium lauroyl sarcosine (sarkosyl), 20 mM HEPES, pH 7.5. The sample was stirred slowly at room temperature for 20 min before centrifugation for 30 min at 204,526 × g. The supernatant was discarded and the pellet was homogenised using a glass dounce in 20 mM Tris, 300 mM NaCl pH 8.0 (TSB buffer) containing 1.5% lauryldimethylamine-N-oxide (LDAO), and incubated at 4 °C with stirring for 1 h. Typically, 100 ml of this buffer was used to extract the OM from 4 l of culture. The sample underwent centrifugation at 204,526 × g for 30 min (45Ti rotor, Beckman) and the supernatant, containing the

extracted OM proteins, was kept. The total protein concentration was determined using a BCA Protein Assay Kit (Thermo Fisher Scientific).

Bt1762-63 was further purified using an Immobilised Metal Affinity Chromatography (IMAC) Ni column using the C-terminal His6 tag on SusD (Bt1762)[15]. The OM fraction from 4 l of MM-Frc grown *B. theta* cells was passed through an ~8–10 ml IMAC Ni column equilibrated with 0.2% LDAO in TSB. The column was washed with TSB with 0.2% LDAO and 25 mM imidazole, and bound proteins were eluted with TSB containing 0.2% LDAO and 200 mM imidazole. IMAC samples were visualised by sodium dodecyl sulfate-polyacrylamide gel electrophoresis (SDS-PAGE). Samples from IMAC were further purified by SEC. An ÄKTA pure system (GE Healthcare) was used in conjunction with a HiLoad 16/60 Superdex 200 pg 120 ml gel filtration column (GE Healthcare). The column was equilibrated with 10 mM HEPES, 100 mM NaCl, 0.05% LDAO, pH 7.5. The IMAC samples were concentrated via centrifugal filtration (Amicon Ultra-15, Millipore; 100 kDa molecular weight cut-off) to 3–5 ml and loaded onto the column. Proteins were eluted at 1.3 ml/min and 3 ml fractions were collected. Peak samples were visualised by SDS-PAGE.

**Crystallisation and structure determination**. For crystallisation, the final SEC purification step was carried out in 10 mM HEPES/100 mM NaCl/0.4% $C_8E_4$, pH 7.5. Fractions were pooled and concentrated to ~15 mg/ml (molecular weight cut-off (MWCO) 100 kDa), aliquoted and flash-frozen in liquid nitrogen. Sitting drop crystallisation trials were set up using a Mosquito crystallisation robot (TTP Labtech) with commercial screens (Molecular Dimensions MemGold 1 and 2 and Morpheus). Optimisation was performed with larger-volume hanging drops set up manually. Well-diffracting crystals for apo-Bt1762-63 were obtained from Mem-Gold 2, condition 18 (0.15 M sodium formate, 0.1 M HEPES pH 7–7.5, 16–20% w/v PEG 3350), and were cryoprotected by the addition of ~20–25% PEG400 for ~5–10 s before being flash-frozen in liquid nitrogen. Diffraction data were collected at 100 K at the Diamond Light Source (Didcot, UK) on beamline i03. Data were processed via Xia2[49] or Dials[50]. The structure was solved by molecular replacement with Phaser[51], using data to 2.62 Å resolution. The previous structure of Bt1762-63 (PDB 5T3R)[15] was used as a search model. The model was built iteratively by manually building in COOT[52], and was refined with Phenix[53] using TLS refinement with 1 group per chain. Structure validation was carried out with Mol-Probity[54]. The model has 94.0% of residues in the favoured regions of the Ramachandran plot, and 0.7% outliers. The structures of Bt1762-63 with bound FOS were obtained by incubation of purified complex with 2.5 mM of the appropriate FOS at room temperature, followed by setting up crystal plates as above. Models of the bound FOS were generated with JLigand within CCP4[55] and fit manually into the density in COOT. Refinement was carried out within Phenix[53] as above. The FOS-bound structures have 90.5/1.0% (PDB ID 6Z9A) and 94.1/0.6% (PDB ID 6ZAZ) of residues in the favourable/disallowed regions of the Ramachandran plot.

**Cryo-EM sample preparation and data collection**. A sample of purified Bt1762-1763 solubilised in an LDAO-containing buffer (10 mM HEPES, pH 7.5, 100 mM NaCl, 0.05% LDAO) was prepared at 0.02 mg/ml. Lacy carbon 300-mesh copper grids coated with a <3 nm continuous carbon film (Agar Scientific) were glow-discharged in air (10 mA, 30 s). A sample volume of 3.5 µl was applied to each grid. Blotting and plunge freezing were carried out using a Vitrobot Mark IV (FEI) with the chamber conditions set at a temperature of 4 °C and 100% relative humidity. A blot force of 6 and a blot time of 6 s were used prior to vitrification in liquid nitrogen-cooled liquid ethane. Micrograph movies were collected on a Titan Krios microscope (Thermo Fisher) operating at 300 kV with a GIF energy filter (Gatan) and K2 Summit direct electron detector (Gatan) operating in counting mode. Data acquisition parameters can be found in Supplementary Table 2.

**Image processing**. Following cryo-EM data collection, the RELION (v3.0) pipeline was used for image processing[56,57]. Drift correction was performed using MotionCor2[58] and contrast transfer functions were estimated using gCTF[59]. Approximately 3000 particles were manually picked, extracted and classified in 2D to provide templates for automated particle picking with RELION. Particles were extracted in 264 × 264 pixel boxes binned to 132 × 132 for initial rounds of 2D and 3D classification. The 3D starting model was generated de novo from the EM data by stochastic gradient descent in RELION. The particle stack was cleaned using both 2D and 3D classification and the remaining particles were unbinned. Further rounds of 3D classification revealed significant conformational heterogeneity in the data. Three predominant conformational states of the levan transport system were identified: OO, OC and CC. Additional classes, while less populated, revealed states with the Bt1762 "lid" open to differing extents (Supplementary Fig. 4). Invariably, the closed position of Bt1762 was associated with an absence of density for the plug domain of Bt1763. Global 3D classification was unable to distinguish "true" closed conformations from those where Bt1762 occupied a marginally open state. As a result, a masked 3D classification approach was employed to achieve homogeneous particle stacks. The masked classification was performed without image alignment and, since the region of interest is relatively small, the regularisation parameter, T, was set to 20. Intermediate results and further details are provided in Supplementary Fig. 13. Clean particle stacks for the three principle conformational states

were subject to multiple rounds of CTF refinement and Bayesian polishing[57]. C2 symmetry was applied to both the OO and CC reconstructions. Post processing was performed using soft masks and yielded reconstructions for the OO, OC and CC states of 3.9, 4.7 and 4.2 Å, respectively, as estimated by gold standard Fourier Shell correlations using the 0.143 criterion.

**Model building into cryo-EM maps**. Comparing the maps to the crystal structure of Bt1762-1763 revealed that their handedness was incorrect. Maps were, therefore, Z-flipped in UCSF Chimera[60]. The reconstruction of the OO state was of sufficient resolution for model building and refinement. Bt1762 and Bt1763 subunits from the crystal structure were independently rigid body fit to the local resolution filtered map and later subjected to several iterations of manual refinement in COOT[52] and "real space refinement" in Phenix[53]. The asymmetric unit was symmetrised in Chimera after each iteration. Molprobity[54] was used for model validation. The reconstructions of the OC and CC states were of insufficient resolution to permit model building and refinement owing to low particle numbers and a poor distribution of viewing angles (CC state). Instead, the crystal structure of Bt1762-1763 was rigid body fit to the CC state. The ligand was removed from the model and an inspection in COOT showed that no density extended past Lys213 in the direction of the N terminus. All residues N terminal of Lys213 were therefore removed from the model before rigid body fitting. The open state from the OO EM structure and the closed state from the crystal structure (modified as described above) were rigid body fit to their corresponding densities in the OC state. Rigid body fitting was performed in Phenix.

**Expression and purification of recombinant enzymes**. Recombinant pET vectors (containing either Bt1760 levanase[19] or *Bacillus levansucrase* genes) were transformed into *E. coli* Tuner cells (Novagen) with appropriate antibiotic selection. One-litre cultures in 2 l flasks were grown to mid-exponential phase at 37 °C in an orbital shaker at 150 r.p.m, then cooled to 16 °C and induced with 1 mM isopropyl β-D-thiogalactopyranoside. These cultures were then incubated for a further 16 h at 16 °C and 150 r.p.m. Cells were harvested by centrifugation, lysed by sonication and the recombinant His-tagged protein purified from cell-free extracts using IMAC (Talon resin, Clontech) as previously described[19].

**Expression and purification of the NTE**. A pBAD24 vector carrying the Bt1763 NTE domain with an N-terminal His6 tag (Supplementary Fig. 6), arabinose promoter and ampicillin resistance was transformed into *E. coli* BL21 (DE3) Lemo cells for expression. Cells were grown in M9 MM containing $^{15}NH_4Cl$ and $^{13}C$-glucose as the source of nitrogen and carbon for protein labelling. Cells were grown at 37 °C for protein expression. At the optical density of 0.7, cells were induced with 0.1% L-arabinose for 3 h. Cells were then harvested by centrifugation at $4400 \times g$ for 15 min, and resuspended in buffer A (20 mM HEPES, 150 mM NaCl, pH 7.5) containing lysozyme, DNAse and protease inhibitor. Homogenised cells were lysed by three rounds passing through a high-pressure microfluidizer. The lysate was centrifuged at $20,000 \times g$ for 30 min at 4 °C. The supernatant was loaded on a Ni-NTA His-trap column (GE Healthcare), washed with 10 column volume of buffer A containing 30 mM imidazole. $^{15}N$, $^{13}C$-labelled NTE was then eluted with three column volumes buffer A containing 200 mM imidazole. Eluted proteins were loaded on Superdex S75 gel filtration column pre-equilibrated with NMR buffer containing 20 mM sodium phosphate, 150 mM NaCl pH 7.5. Fractions corresponding to pure NTE were concentrated through a 5 kDa cut-off filter for NMR spectra acquisitions. The NMR sample contained 250 µl 1 mM U-$^{15}N$, $^{13}C$-NTE in NMR buffer containing 5% $^2H_2O$.

**NMR experiments**. All NMR spectra were recorded at 20 °C on a Bruker Avance-700 spectrometer equipped with a cryogenic triple-resonance probe and using Topspin 3.6. The proton chemical shifts were referenced to water, and $^{15}N$ and $^{13}C$ were indirectly referenced. 2D [$^{15}N$,$^1H$]-HSQC, 2D [$^{13}C$,$^1H$]-HSQC, 3D HNCACB, 3D CBCA(CO)NH, 3D $^{15}N$-resolved-[$^1H$,$^1H$]-NOESY and 3D $^{13}C$-resolved-[$^1H$,$^1H$]-NOESY were acquired for backbone resonance assignment and structure determination of NTE. The NMR experiments and respective acquisition parameters are listed in Supplementary Tables 3 and 4.

**Calculation of the 3D solution structure of NTE**. The side-chain chemical shifts were assigned automatically using FLYA algorithm with fixed backbone chemical shifts[61]. $^1H$-$^{15}N$ and $^1H$-$^{13}C$ NOE cross-peaks were automatically assigned followed by manual corrections, resulting in a total of 964 unambiguously assigned peaks. Backbone torsion angle constraints were derived from chemical shift values using the program TALOS[62]. Using these values, the solution structure calculation was performed using the program CYANA[63].

**FOS production and purification**. Two grams of *E. herbicola* levan (Sigma-Aldrich) were dissolved in 100 ml phosphate-buffered saline (PBS) (Oxoid) by autoclaving. The solution was cooled to room temperature before the addition of 100 nM Bt1760 GH32 endo-acting levanase and incubated at 37 °C for 30 min to partially digest the levan to a mixture of different size β2,6 FOS. The enzyme was then heat inactivated by boiling for 20 min. The resultant sample was freeze dried

using a Christ Alpha 1-2 Freeze Drier at −45 °C. The freeze-dried FOS mixture was resuspended in 5 ml 50 mM acetic acid and loaded onto a column (two $2.5 \times 80\, cm^2$ Glass Econo-Columns, connected in series with a flow adaptor; Bio-Rad) packed with P2 Bio-gel size-exclusion resin (Bio-Rad) and pre-equilibrated with 50 mM acetic acid. The column was run at 0.25 ml/min using a peristaltic pump (LKB Bromma 2132 microperpex) and 2 ml fractions were collected continuously for 48 h using a Bio-Rad model 2110 fraction collector. Fractions were initially analysed by TLC and any that contained sugar were freeze dried to remove acetic acid.

**Thin-layer chromatography**. TLC plates (Silica gel 60, Sigma-Aldrich) were cut to the required size, and samples were spotted 1 cm from the bottom of the plate. These were dried using a hair dryer and placed into a tank containing 1 cm of running buffer (1-butanol, acetic acid and water at 2:1:1). Once the running buffer migrated to 1 cm from the top of the plate, the plate was then dried and run again. To visualise sugars, the plates were completely dried and submerged in developer solution (sulphuric acid, ethanol and water at 3:70:20 with 1% orcinol) for 5–10 s. Finally, the plate was dried using a hair dryer and incubated at 65 °C in a drying oven until developed (usually ~1 h).

**Isothermal titration calorimetry**. SEC-purified Bt1762-63-SusCD complex was dialysed overnight at 4 °C with 50 kDa MWCO dialysis tubing into 100 mM HEPES, pH 7.5, containing 0.05% LDAO. The dialysis buffer was used to resuspend SEC-purified FOS fractions produced by partial digestion of *Erwinia* levan. ITC was performed using a MicroCal PEAQ-ITC machine with v1.21 control software for data collection (Malvern Panalytical). Briefly, pure SusCD complex (25–50 μM; concentration determined by $A_{280}$) in a 200 μl reaction well was injected 20 times with 2 μl aliquots of FOS (0.25–4.5 mg/ml) at 25 °C. Integrated heats were fit to a one set of sites model using the Microcal PEAQ-ITC analysis software v1.30 (Malvern Panalytical) to obtain $K_d$, DH and N (number of binding sites on the protein). For most titrations, the molar concentration of ligand used for the fits was based on the DP of the major FOS species present as determined by the TLC and MS analysis. For fractions 118–115, the concentration of ligand was varied such that $N = 1$.

**Native MS**. Proteins were buffer exchanged into 0.5% $C_8E_4$, 0.2 M ammonium acetate, pH 6.9, using a micro biospin 6 column (Bio-Rad). The FOS sample was diluted to ~500 μM with 0.5 M ammonium acetate, pH 6.9. Mass spectra were acquired on a Q-Exactive hybrid quadrupole-Orbitrap mass spectrometer (Thermo Fisher Scientific, Bremen, Germany) optimised for transmission and detection of high molecular weight protein complexes. About 3 μl of aliquot of the sample was transferred into gold-coated borosilicate capillary (Harvard Apparatus) prepared in-house and mounted on the nano-electrospray ionization source. The instrument settings were 1.2 kV capillary voltage, S-lens RF 200%, argon ultra-high vacuum pressure $3.1 \times 10^{-10}$ mbar, capillary temperature 100 °C. Voltages of the ion transfer optics–injection flatapole, inter-flatapole lens, bent flatapole, and transfer multipole were set to 5, 3, 2, and 30 V, respectively. The noise level was set at 3. Protein ions were activated with −120 V with the in-source trapping mode and a collisional activation voltage of 300 V. Data were visualised and exported for processing using the Qual browser of Xcalibur 4.2 (Thermo Scientific).

**Growth curves**. Growth curves were performed in an Epoch microplate spectrometer (Biotek Instruments Ltd) with 96-well Costar culture plate (Sigma-Aldrich) inside an anaerobic cabinet at 37 °C (Don Whitley Scientific, A35 workstation). Media were inoculated 1:10 with bacterial cultures previously grown overnight in BHI. Final culture volumes of 200 μl were used, and each condition was performed in triplicate. Optical density at 600 nm was measured in each well at 30 min intervals.

**Sources of levan used**. In most cases, the levan used was from *Erwinia herbicola* (Sigma). For analysis of the growth of the Δ1760 strain, bacterial levans from *Bacillus* sp. (Montana Polysaccharides) and *Zymomonas mobilis*[64] (a kind gift from Prof. Dr. Yekta Göksunger, Ege University, Izmir, Turkey) were also used, as well as levan from Timothy grass (Megazyme) and in vitro synthesised levan. In vitro synthesised levan was made using *Bacillus subtilis* levansucrase incubated with 20% sucrose in PBS for 24 h at 37 °C. Proteinase K (100 μg/ml) was then added to remove the protein, before being precipitated out using 0.5% trichloroacetic acid final. The levan was then extensively dialysed against water and freeze dried.

**Reporting summary**. Further information on research design is available in the Nature Research Reporting Summary linked to this article.

## Data availability
The data supporting the findings of this study are available from the corresponding authors upon reasonable request. Coordinates and structure factors that support the findings of this study have been deposited in the Protein Data Bank with accession codes 6ZAZ (Bt1762-63 with shorter FOS), 6Z8I (Bt1762-63 apo) and 6Z9A (Bt1762-63 with longer FOS). EM structure coordinates have been deposited in the Electron Microscopy Data Bank with accession codes 6ZLT (OO; EMD-11273), 6ZM1 (OC; EMD-11277) and 6ZLU (CC; EMD-11274). The NMR data have been deposited and publicly released in

the Biological Magnetic Resonance Bank (BMRB) with number 34514. The NTE structure has been deposited in the Protein Data Bank with accession code 6YTC. Source data are provided with this paper.

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

## Acknowledgements

D.A.G., D.N.B and B.v.d.B. were supported by a Biotechnology and Biological Sciences Research Council grant (BB/P003192/1). J.B.R.W. was supported by a Wellcome Trust 4-year Ph.D. studentship (215064/Z/18/Z). All electron microscopy was performed at the Astbury Biostructure Laboratory, which was funded by the University of Leeds and the Wellcome Trust (108466/Z/15/Z). We thank R. Thompson, E. Hesketh and D. Maskell for electron microscopy support. S.H. was supported by the Swiss National Science Foundation (grant 167125). A.O.O. and C.V.R. are supported by a Medical Research Council grant (MR/N020413/1). We would further like to thank the personnel of the Diamond Light Source beamlines i03, i04 and i04-1 for beamtime (Block Allocation Group numbers mx-13587 and mx-18598) and assistance with data collection.

## Author contributions

D.A.G made *B. theta* mutants, expressed and purified proteins, performed growth assays, generated FOS substrates and carried out ITC. J.B.R.W. and S.L.E. determined cryo-EM structures, supervised by N.A.R. A.O.O. carried out native mass spectrometry, supervised by C.V.R. P.R., A.M. and M.Z. determined the NTE structure, supervised by S.H. A.J.G. contributed to all early stages of the project. C.M. made *B. theta* mutants. A.B. collected X-ray crystallography data. A.C. helped analyse crystallography data. B.v.d.B. crystallised proteins and determined crystal structures. The manuscript was written by B.v.d.B., D.N.B. and D.A.G., with input from J.B.R.W., N.R., C.V.R. and S.H.

## Competing interests

The authors declare no competing interests.
