## [Peer Review File · Nature Communications]

REVIEWER COMMENTS

Reviewer #1 (Remarks to the Author):

This is a beautiful manuscript that combines a number of structural and biophysical techniques to probe glycan import across the outer membrane of an important gut bacterium. Crystal structures reveal residues involved in substrate recognition and suggest that the unusually large binding cavity can accommodate several substrates of 2.5 kDa each, a finding supported by ITC and native mass spec experiments. Cryo-EM of the SusCD complex reveals several states of the transporter, allowing us to visualize the pedal bin transport mechanism proposed by van den Berg and coworkers several years ago.

Minor points to clarify:

1. There is one particular experiment presented that could use some more discussion, or the authors could at least comment and suggest a possible explanation. In figure 4b the authors show the effect of hinge deletions on bacterial growth. Deletion of hinge 1 caused little effect, while deletion of hinge 2 abolished growth and deletion of both hinges resulted in growth similar to that seen in the mutSusD strain. Though the authors comment that this was quite surprising (they expected growth to be abolished) no discussion or suggestions for why this may be happening were given.
2. Toward the end of the paper (lines 489-491), the authors mention the possible role of SGBP in substrate delivery (replacing SusD). If these proteins are in close association to SusC, this could explain why growth similar to WT is seen in the mutSusD. However, this still doesn't explain why growth would be abolished in a hinge 2 deletion but not deletion of both hinges.
3. In addition, the authors should include the resolution of each structure in the figure 1b legend.
4. Page 7 line 184-185: Introduce as OO, OC and CC in the text to match presentation of structures in figure 3a.

Reviewer #2 (Remarks to the Author):

In this work, Gray and colleagues report crystal structures of the β 2,6 fructo-oligosaccharide (FOS) importing SusCD from *Bacteroides thetaiotaomicron* (Bt1762-Bt1763), in (i) its unliganded form, and in complex with (ii) DP15-25 and (iii) DP6-12, at 2.62, 2.99 and 2.69 Å resolution, respectively. The structures reveal the residues involved in FOS recognition and identify a cavity likely implicated in the recognition of substrate molecules up to ca. 2.5 kDa in size. These findings are supported by native mass spectrometry, isothermal titration calorimetry, and mutational studies in vivo. The authors previously reported two functionally distinct SusCD complexes purified from *Bacteroides thetaiotaomicron*, including the unliganded form of Bt1762-Bt1763, and proposed the 'pedal bin' mechanism as a general model for substrate translocation (reference 15). In this manuscript, Cryo-EM of the intact SusC2D2 complex reveals several distinct states of the transporter, including an open-open, open-closed, and closed-closed, supporting the proposed mechanism. The quality of the data is very high. The manuscript is clear and well written.

I support the publication of this manuscript in Nature Communications.

Please find below, some comments/suggestions for improvement:

1. Figures

- Figure 1: Please, introduce a new panel showing (i) the dimeric nature of the Bt1762-Bt1763 complex, and (ii) the location of FOS1 and FOS2 molecules in both protomers.
- Figure 2: Please, provide new panels showing (i) the location of the FOS1 and FOS2 substrates in a selected, surface calculated area of the Bt1762-Bt1763 complex, and (ii) the electrostatic surface potential of the ligand-binding cavity.
- "Like the oligopeptide ligands in the RagAB and Bt2261-64 structures, the FOS is bound at the top of a large, solvent-excluded cavity formed by the Bt1762-63 complex." A good opportunity to prepare a Figure showing the corresponding structural comparison.
- Figure 3: Please, increase the size of each panel.
- Please, move Extended Data Figures 2 and 3 to the main text, to better reflect the proposed mechanism.

2. Tables

- Table 1: Please include the following data:

1. Wavelength (Å)
2. Total reflections (with the highest-resolution shell shown in parentheses).
3. Unique reflections (with the highest-resolution shell shown in parentheses).
4. Wilson B-factor.
5. R-merge.
6. R-meas.
7. CC1/2.
8. Reflections used in refinement.
9. Reflections used for R-free.
10. CC(work).
11. CC(free).
12. Ramachandran favored (%).
13. Ramachandran outliers (%).

Reviewer #3 (Remarks to the Author):

Title: "Insights into glycan import by a prominent gut symbiont."

Authors: Declan A. Gray, Joshua B. R. White, Abraham O. Oluwole, Parthasarathi Rath, Amy J. Glenwright, Adam Mazur, Michael Zahn, Arnaud Baslé, Carl Morland, Sasha L. Evans, Alan Cartmell, Carol V. Robinson, Sebastian Hiller, Neil A. Ranson, David N. Bolam and Bert van den Berg

In this study the authors present structural and biophysical data for the SusCD transport complex from *Bacteroides thetaiotaomicron* (Bt1762-Bt1763). They present the X-ray crystal structure of the complex in the presence of different length oligosaccharides to gain insights into specificity of the length of the transport cargo. The size and specificity of the cargo is further evaluated using Mass Spec and ITC binding assays. Additionally, they are able to provide cryoEM data showing potential transporter states of the complex. The authors also provide structural information on a unique NTE

domain of SusC, whose function is unclear. Growth and binding characteristics for different mutant transporters and different size cargo are also presented. Taken together the presented data support the pedal bin hypothesis of cargo recognition and transport for this type of TonB-dependent transporter. While the manuscript has merit, there are issues with the cohesion of the manuscript. This is especially true in the Materials and Methods where most descriptions are appropriate, but others become very vague and insufficient for actual reproduction of the experiments. It is also true in some of the language used in the results, for example the authors mention normal occupancy. What is normal occupancy? Is it the mostly missing occupancy of previous structures or 1.0 where the atoms are present? Additionally, they have a tendency to over state their findings, such as when they state they have unambiguously assigned the saccharide at 3.1 Å resolution, but not provided sufficient explanation of how all possible conformers were modelled as well as when they insist a 4-mer is present while stating a 3-mer fits better. They also state that the FoxA NTE is similar to their NTE with substitutions of alpha helix in FoxA for beta structure in their NTE as presented in Fig 5. This is clearly not the case as the connectivity is drastically different and trace of the peptide chain is completely different. From a structural point of view this is completely incorrect. At the very least the authors need to reword their case because the presented figures do not do their argument justification. However, overall this manuscript is a real tour de force to understand the SusCD transport mechanism and that has merit.

In general the paper is well written and clear in its presentation. However, there are certain aspects of the paper that can be improved. It is often confusing why the authors sometimes refer to the SusCD complex as Bt1762-1763 and subsequently other proteins as the Bt#### versus their gene product name. It is suggested that the authors possibly add a table to clearly delineate genomic loci with presumed protein names (Bt 1762-1763, SusCD), this could streamline their presentation especially where other gene products are concerned. It is recognized that they have tried to present an operon overview, but not all the gene products are described. Color coding was helpful, but there is some color overlap and missing representations, which resulted in confusion for this reviewer. A more detailed list of points to address, which are mostly minor follows:

- 1) Ln 77: Glycan binding proteins is confusing because the glycan binding occurs for different reasons. SusD (Bt 1762) is glycan binding because it is used for levan fragment recognition. However, Bt 1761 is used to bind presumably larger forms of levan near the endo-levanase Bt 1760. While technically true they both bind levane (since it is presumably polydisperse) and the processing of levane is necessary for recognition by Bt 1762, it is suggested this be clarified.
- 2) Fig 1.: The following points are confusing to this reviewer:
 - a. Is Bt 1759 a beta-fructosidase? What is the function of Bt 1765? Do these both have the same function? They are colored the same and the beta-fructosidase doesn't have a Bt label in the figure.
 - b. Bt 1754 seems to be an undefined PUL sensor that is not represented in the cartoon. PUL sensing is not represented.
 - c. Is Bt 1757/58 an ABC transporter as is typical for these systems? The current representation of Bt 1758 is cylindrical, presumably because it is a beta-barrel transporter, which is not common for the inner membrane. Bt1757 isn't even shown. This should be clarified.
- 3) Need more information about the dynamics of the plug for Extended Data Fig 1. Why are you making a conclusion about movement of the plug within the barrel vs a conclusion that most of the plug is not present? More evidence is needed.
- 4) Ln 119: The plug is not shown in the left panel of Fig. 1.
- 5) Ln 127: If the plug is often ejected, what is the relevance of the NTE? How does substrate stabilization affect the location of the NTE? Is this information available in the cryoEM structures?
- 6) Ln 132: What is normal occupancy? 1.0 since the peptide backbone is presumably always there or 0.0 since the plug is usually missing. The wording needs to be more accurate.
- 7) Ln 136: How are the authors able to unambiguously assign the 7-mer? How many different models were tried of 6-mers or 5-mers with variable conformations? At 3.1 Å it is hard to believe there is no ambiguity. A 7-mer may have best fit the density, but it is hardly unambiguous.
- 8) Ln 138: The authors comment that the conformation of the sugar is helical. A comment on how this compares to known saccharide structures would be helpful.

- 9) Ln 147: the resolution comment is clunky. Consider a revision.
- 10) Fig. 2: Align the labels of the figure.
- 11) Ln 154: Can the fit for the 3-mer be shown in comparison to the 4-mer? It is stated the 3-mer fits better. Why is the 4-mer preferred?
- 12) Ln 160-1: It is assumed that modelling was attempted. It should be reported about the poor fit. This is presumed what is meant when the authors state low resolution didn't allow building.
- 13) Ln 176: crystallization or crystallization. This reviewer prefers crystallizations.
- 14) Ln 182: Do the authors see the lipid anchor? If so, a comment here would be useful.
- 15) Ln 200: What is the overall RMSD? A backbone deviation of 2 – 2.5 Å is significant. It is suggested this is moved to the main manuscript.
- 16) Ln 239: What is the concentration of the 0.5 % levan in molarity? How does this compare to the concentration tested by ITC?
- 17) Ln 322: References for all structures of STN domains should be made.
- 18) Ln 325: The recent complex structure of an STN with a sigma regulator should be referenced here. J Biol Chem. 2020 Apr 24;295(17):5795-5806. doi: 10.1074/jbc.RA119.010697.
- 19) Ln 331: There is no similarity of the FoxA STN to the NTE. This needs to be made more clear from a proper protein structure point of view.
- 20) Ln. 343: "in several Sus" does not make sense. Please expand.
- 21) Ln 349: DP is undefined.
- 22) Ln 370: What is the evidence for beta 2,1 decorations? Is there another Levan that has more?
- 23) Ln 375: What is the concentration of the FOS in the media? Is it possible that it is being pushed through on a concentration gradient, which would make the binding unnecessary? This possibility should probably be mentioned.
- 24) Ln 397: Why not perform the ITC at 6-fold concentration to see binding? This would remove ambiguity.
- 25) Ln 411-422: Why not grow the cultures on media that has been depleted of smaller fragments by growth of the mutant? Essentially grow the mutant twice on the same media after removal of the bacteria.
- 26) Fig. 7. Is confusing. In 7a the +33 peak is labeled, but in 7b the +32 peak is labeled. Why did the authors switch from +33 to +32? Just label the +32 peak to avoid confusion.
- 27) Ln 490: It seems that a concentration gradient could also explain this.
- 28) Ln 587: FUdR is undefined.
- 29) Ln 593: BHI is defined, do not spell out.
- 30) Ln 595: MM is already defined.
- 31) Ln 597: TSB is not defined.
- 32) Ln 606: What is the concentration of LDAO? It is not defined.
- 33) Ln 622: Buffer exchange is not detailed. This is one of the places where the presentation of the detail in the methods is incongruous. The authors should make sure that sufficient detail is provided to reproduce the experiments. That detail is lacking here.
- 34) Ln 714: Presumably the authors used U-15N and U-13C to obtain uniformly labeled protein for the NMR experiments.
- 35) Ln 812: The authors have not reported the NMR data deposition. The data should be deposited at the BMRB - Biological Magnetic Resonance Bank and reported here with the other data deposition information. This needs to be done prior to acceptance.
- 36) Extended Fig 1: This figure should be remade. The cartoon representation is not useful when trying to determine the quality of electron density as it is designed to be a smoothed representation of the backbone rather than an accurate description. The authors should either use a true Alpha backbone representation or a full atom "sticks" representation to show how the quality of the electron density.
- 37) Supplementary Table 1: The $I/\sigma I$ for the outermost shell is quite low. It is presumed that the authors are using a CC1/2 cutoff to define the $I/\sigma I$ cutoff. This CC1/2 value should be reported for the outermost shell or the resolutions should be cut back to $I/\sigma I = 2.0$. If the CC1/2 was used then reporting this would show that the data with $I/\sigma I < 2$ does actually have useful information. Otherwise it is an erroneous assumption that data that is poorly measured contributes positively to the model.

Hopefully the authors performed a paired refinement to get a true CC1/2. If they haven't, this is trivial to do in Phenix and should be done prior to acceptance.

REVIEWER COMMENTS

Reviewer #1 (Remarks to the Author):

This is a beautiful manuscript that combines a number of structural and biophysical techniques to probe glycan import across the outer membrane of an important gut bacterium. Crystal structures reveal residues involved in substrate recognition and suggest that the unusually large binding cavity can accommodate several substrates of 2.5 KDa each, a finding supported by ITC and native mass spec experiments. Cryo-EM of the SusCD complex reveals several states of the transporter, allowing us to visualize the pedal bin transport mechanism proposed by van den Berg and coworkers several years ago.

Minor points to clarify:

1. There is one particular experiment presented that could use some more discussion, or the authors could at least comment and suggest a possible explanation. In figure 4b the authors show the effect of hinge deletions on bacterial growth. Deletion of hinge 1 caused little effect, while deletion of hinge 2 abolished growth and deletion of both hinges resulted in growth similar to that seen in the mutSusD strain. Though the authors comment that this was quite surprising (they expected growth to be abolished) no discussion or suggestions for why this may be happening were given.

It is true that this is an unexpected result and we currently have no clear explanation for the observed phenotype for the double hinge mutant. The level of SusCD expression is similar in the hinge 2 and double mutant so we don't think it is related to this. It may be that the loss of both hinges from SusC allows too much freedom to the SusD lid, which will only be attached to the SusC via the SusD N-terminal ~15aa extension following the lipid anchor. This freedom may essentially decouple substrate binding by SusD from delivery to SusC, generating a phenotype similar to that of delta SusD. The complete loss of growth observed for the hinge 2 mutant could possibly be due to a "dominant negative" effect by something that does not involve the levan PUL but is essential for growth, perhaps due to SusC misfolding that leads to stalling of the BamA assembly machine. These are obviously simply speculations as we have no experimental evidence for either, but we have added this possible explanation to the text, lines 278-287.

2. Toward the end of the paper (lines 489-491), the authors mention the possible role of SGBP in substrate delivery (replacing SusD). If these proteins are in close association to SusC, this could explain why growth similar to WT is seen in the mutSusD. However, this

still doesn't explain why growth would be abolished in a hinge 2 deletion but not deletion of both hinges.

As mentioned above, we speculate that in the hinge 2 mutant (as well as in the Δ NTE mutant) there is a "dominant negative" effect caused by something (not necessarily the same something) that does not involve the levan PUL but is essential for growth.

3. In addition, the authors should include the resolution of each structure in the figure 1b legend.

Resolutions are now included as requested (this is now Fig. 1d).

4. Page 7 line 184-185: Introduce as OO, OC and CC in the text to match presentation of structures in figure 3a.

Change made as requested.

Reviewer #2 (Remarks to the Author):

In this work, Gray and colleagues report crystal structures of the β 2,6 fructo-oligosaccharide (FOS) importing SusCD from *Bacteroides thetaiotaomicron* (Bt1762-Bt1763), in (i) its unliganded form, and in complex with (ii) DP15-25 and (iii) DP6-12, at 2.62, 2.99 and 2.69 Å resolution, respectively. The structures reveal the residues involved in FOS recognition and identify a cavity likely implicated in the recognition of substrate molecules up to ca. 2.5 kDa in size. These findings are supported by native mass spectrometry, isothermal titration calorimetry, and mutational studies in vivo. The authors previously reported two functionally distinct SusCD complexes purified from *Bacteroides thetaiotaomicron*, including the unliganded form of Bt1762-Bt1763, and proposed the 'pedal bin' mechanism as a general model for substrate translocation (reference 15). In this manuscript, Cryo-EM of the intact SusC2D2 complex reveals several distinct states of the transporter, including an open-open, open-closed, and closed-closed, supporting the proposed mechanism. The quality of the data is very high. The manuscript is clear and well written.

I support the publication of this manuscript in Nature Communications.

Please find below, some comments/suggestions for improvement:

1. Figures

- Figure 1: Please, introduce a new panel showing (i) the dimeric nature of the Bt1762-Bt1763 complex, and (ii) the location of FOS1 and FOS2 molecules in both protomers.

This new panel has been included as Figure 1c.

- Figure 2: Please, provide new panels showing (i) the location of the FOS1 and FOS2 substrates in a selected, surface calculated area of the Bt1762-Bt1763 complex, and (ii) the electrostatic surface potential of the ligand-binding cavity.

We have not included the requested panels because we do not feel they are very informative and notably do not add to what is shown already in Figs. 1 and 2 (note that the sugars are uncharged).

- “Like the oligopeptide ligands in the RagAB and Bt2261-64 structures, the FOS is bound at the top of a large, solvent-excluded cavity formed by the Bt1762-63 complex.” A good opportunity to prepare a Figure showing the corresponding structural comparison.

We agree and we have now made a new SI Figure showing this (SI Fig. 1). Please note that this has changed the numbering of all other SI Figures in the revision.

- Figure 3: Please, increase the size of each panel.

Done as requested. Please note we have now combined the original ED Fig. 3 panel b with this Figure. The original Fig. 3b-d have now replaced the original ED Fig. 3 to made the new SI Fig. 3.

- Please, move Extended Data Figures 2 and 3 to the main text, to better reflect the proposed mechanism.

We have opted not to move ED Fig. 2 (now SI Fig. 4) to the main text, since we think it is better suited for the SI section. ED Fig. 3 (panel b) has now been combined with Figure 3.

2. Tables

- Table 1: Please include the following data:

1. Wavelength (Å)
2. Total reflections (with the highest-resolution shell shown in parentheses).
3. Unique reflections (with the highest-resolution shell shown in parentheses).
4. Wilson B-factor.
5. R-merge.
6. R-meas.
7. CC1/2.
8. Reflections used in refinement.
9. Reflections used for R-free.
10. CC(work).
11. CC(free).
12. Ramachandran favored (%).
13. Ramachandran outliers (%).

We have included some of the requested numbers, but not the infrequently used ones.

Reviewer #3 (Remarks to the Author):

Title: "Insights into glycan import by a prominent gut symbiont."

Authors: Declan A. Gray, Joshua B. R. White, Abraham O. Oluwole, Parthasarathi Rath, Amy J. Glenwright, Adam Mazur, Michael Zahn, Arnaud Baslé, Carl Morland, Sasha L. Evans, Alan Cartmell, Carol V. Robinson, Sebastian Hiller, Neil A. Ranson, David N. Bolam and Bert van den Berg

In this study the authors present structural and biophysical data for the SusCD transport complex from *Bacteroides thetaiotaomicron* (Bt1762-Bt1763). They present the X-ray crystal structure of the complex in the presence of different length oligosaccharides to gain insights into specificity of the length of the transport cargo. The size and specificity of the cargo is further evaluated using Mass Spec and ITC binding assays. Additionally, they are able to provide cryoEM data showing potential transporter states of the complex. The authors also provide structural information on a unique NTE domain of SusC, whose function is unclear. Growth and binding characteristics for different mutant transporters and different size cargo are also presented. Taken together the presented data support the pedal bin hypothesis of cargo recognition and transport for this type of TonB-dependent transporter. While the manuscript has merit, there are issues with the cohesion of the manuscript. This is especially true in the Materials and Methods where most descriptions are appropriate, but others become very vague and insufficient for actual reproduction of the experiments. It is also true in some of the language used in the results, for example the authors mention normal occupancy. What is normal occupancy? Is it the mostly missing occupancy of previous structures or 1.0 where the atoms are present? Additionally, they have a tendency to overstate their findings, such as when they state they have unambiguously assigned the saccharide at 3.1 Å resolution, but not provided sufficient explanation of how all possible conformers were modelled as well as when they insist a 4-mer is present while stating a 3-mer fits better. They also state that the FoxA NTE is similar to their NTE with substitutions of alpha helix in FoxA for beta structure in their NTE as presented in Fig 5. This is clearly not the case as the connectivity is drastically different and trace of the peptide chain is completely different. From a structural point of view this is completely incorrect. At the very least the authors need to reword their case because the presented figures do not do their argument justification. However, overall this manuscript is a real tour de force to understand the SusCD transport mechanism and that has merit.

In general the paper is well written and clear in its presentation. However, there are certain aspects of the paper that can be improved. It is often confusing why the authors sometimes refer to the SusCD complex as Bt1762-1763 and subsequently other proteins as the Bt##### versus their gene product name. It is suggested that the authors possibly add a table to clearly delineate genomic loci with presumed protein names (Bt 1762-1763, SusCD), this could streamline their presentation especially where other gene products are concerned. It is recognized that they have tried to present an operon overview, but not all the gene products are described. Color coding was helpful, but there is some color overlap and missing representations, which resulted in confusion for this reviewer. A more detailed list of points to address, which are mostly minor follows:

- 1) Ln 77: Glycan binding proteins is confusing because the glycan binding occurs for different reasons. SusD (Bt 1762) is glycan binding because it is used for levan fragment recognition. However, Bt 1761 is used to bind presumably larger forms of levan near the

endo-levanase Bt 1760. While technically true they both bind levane (since it is presumably polydisperse) and the processing of levane is necessary for recognition by Bt 1762, it is suggested this be clarified.

This has now been clarified in the text, line 77.

2) Fig 1.: The following points are confusing to this reviewer:

a. Is Bt 1759 a beta-fructosidase? What is the function of Bt 1765? Do these both have the same function? They are colored the same and the beta-fructosidase doesn't have a Bt label in the figure.

Both BT1759 and BT1765 are GH32 beta-fructosidases that remove fructose from imported levan-oligosaccharides in the periplasm. This has now been clarified in the Fig. 1 legend.

b. Bt 1754 seems to be an undefined PUL sensor that is not represented in the cartoon. PUL sensing is not represented.

BT1754 is a hybrid two component system that activates PUL expression in response to fructose in the periplasm (Sonnenburg et al 2010 - Ref #19). This is now stated in the legend to Fig 1. The role of BT1754 is also mentioned in the Results section, line 100.

c. Is Bt 1757/58 an ABC transporter as is typical for these systems? The current representation of Bt 1758 is cylindrical, presumably because it is a beta-barrel transporter, which is not common for the inner membrane. Bt1757 isn't even shown. This should be clarified.

We agree this is not clear and we have modified text to clarify. BT1758 is a predicted MFS transporter and therefore likely imports released fructose from the periplasm to the cytoplasm. BT1757 is predicted to be a fructokinase, but the role for this protein in fructan metabolism is not clear. The close association of the BT1757/BT1758 genes suggests they have related functions but this is currently only speculation. We have redrawn BT1758 in Fig. 1a to resemble an MFS transporter and described its predicted function in the legend and panel 1b to remove ambiguity, as well as the predicted function of BT1757.

3) Need more information about the dynamics of the plug for Extended Data Fig 1. Why are you making a conclusion about movement of the plug within the barrel vs a conclusion that most of the plug is not present? More evidence is needed.

We have rewritten this. Density for the plug is poor but clearly present, suggesting mobility or different conformations of the plug within the barrel. Please see lines 161-167.

4) Ln 119: The plug is not shown in the left panel of Fig. 1.

We have not shown it because the plug is not present in its native conformation. The poor density does not allow building of a model for the plug (as indicated in SI Fig. 2). Please note that this panel is now the middle panel of Fig. 1e.

5) Ln 127: If the plug is often ejected, what is the relevance of the NTE? How does substrate stabilization affect the location of the NTE? Is this information available in the cryoEM structures?

We assume the reviewer was referring to line 122 here, not 127. The possible ejection of the plug is most likely a consequence of removing the protein from the lipid bilayer, *i.e.* it is not something that will happen spontaneously, without TonB interaction, *in vivo*. We're not sure what is meant with "relevance of the NTE". As discussed in the manuscript, we do not know what the function of the NTE is. There is only very weak density for the NTE in the open states of the cryo-EM structures.

6) Ln 132: What is normal occupancy? 1.0 since the peptide backbone is presumably always there or 0.0 since the plug is usually missing. The wording needs to be more accurate.

We agree this is confusing and have removed the statement.

7) Ln 136: How are the authors able to unambiguously assign the 7-mer? How many different models were tried of 6-mers or 5-mers with variable conformations? At 3.1 Å it is hard to believe there is no ambiguity. A 7-mer may have best fit the density, but it is hardly unambiguous.

This issue is probably related to the way we wrote the original manuscript, where we discussed first the apo structure, followed by the 3.1 Å co-crystal structure, and ending with the 2.69 Å co-crystal structure. However, the order in which the structures were determined was the 2.69 Å co-crystal structure first, followed by apo, and finally the 3.1 Å co-crystal structure. We have now rewritten the manuscript to reflect the chronological order of the experiments. Thus, the sugar fitting was done in the higher resolution structure, and the lower-resolution structure determined later simply had very similar density that made "fitting" of the same sugar model easy. We're not sure what the reviewer means by "variable conformations". The structure of levan is known (b2-6 linked fructose units with b2-1 branches; lines 72-74), so fitting the sugars to the density was relatively simple. We do agree that "unambiguous" might not be justified even at 2.69 Å resolution, so we have rephrased this (line 132). Please note that the order of the panels in Fig. 1e is now different to reflect these changes in the text.

8) Ln 138: The authors comment that the conformation of the sugar is helical. A comment on how this compares to known saccharide structures would be helpful.

The helical structure of the beta-2,6 FOS bound to Bt1762-63 is very similar to the conformation of the levan-tetraose bound to BT1760 endo-levanase (PDB 6R3U Ernits et al 2019 - new ref #26). This is the only other beta-2,6 configured fructan structure in the PDB. This is now stated in the main text line 135.

9) Ln 147: the resolution comment is clunky. Consider a revision.

We're sorry, but we do not agree with this remark, assuming that "clunky" refers to the phrase "using data to 3.1 Å". Our wording is appropriate because it is not possible to refer to "the resolution" of a dataset. The data is always cut off according to certain criteria (e.g. CC1/2 of 0.3). Thus, "the resolution of the data" is a choice and not something intrinsic to the data.

10) Fig. 2: Align the labels of the figure.

Labels aligned as requested.

11) Ln 154: Can the fit for the 3-mer be shown in comparison to the 4-mer? It is stated the 3-mer fits better. Why is the 4-mer preferred?

It is very hard to show such differences in "fit quality" (which are small and to some extent arbitrary) in static pictures. We feel that interested readers would be better served by inspecting the fit to the density for different sugar models themselves. We now say that the fit to the FOS2 density "is slightly better" for a branched 3-mer compared to a "straight" 4-mer (line 147). We feel we are sufficiently cautious with our suggestion that the system "may have some specificity for FOS with a β 2,1 decoration" (line 142).

12) Ln 160-1: It is assumed that modelling was attempted. It should be reported about the poor fit. This is presumed what is meant when the authors state low resolution didn't allow building.

The density is too poor to allow modelling, and the term "fit" is not appropriate here. We do not quite understand what the problem is with this text (now lines 174-176), and have therefore left it unchanged.

13) Ln 176: crystallization or crystallization. This reviewer prefers crystallizations.

We respectfully disagree. The process of crystallisation selects for compact states.

14) Ln 182: Do the authors see the lipid anchor? If so, a comment here would be useful.

We assume the reviewer meant line 187? Density for the lipid anchors (i.e. the acyl chains) is present in some maps but very weak. We have opted not to build them and feel a comment would detract from the flow of the text.

15) Ln 200: What is the overall RMSD? A backbone deviation of 2 – 2.5 Å is significant. It is suggested this is moved to the main manuscript.

The overall RMSD is quite low (~ 0.5 Å for the C alpha carbons), since much of the structures are identical. We think it is better to keep the Figure in the SI.

16) Ln 239: What is the concentration of the 0.5 % levan in molarity? How does this compare to the concentration tested by ITC?

W85A vs DP9 FOS by ITC used 0.33 mg/ml of pure DP9 (same as for WT - this is the highest affinity ligand), while levan in the media is at 5 mg/ml - so 15x higher weight of sugar. However, levan is a heterogeneous mix of FOS so the effective molar concentration of levan is not known. Notably, levan is mainly composed of FOS that is larger than DP9 (see newly added TLC of undigested levan Supplementary Fig 8 - which shows levan remains at the origin) and thus DP9 will only be a very small fraction of the total FOS in the media, and very likely at a considerably lower concentration than that used for ITC.

17) Ln 322: References for all structures of STN domains should be made.

Included as requested for the STN domains of FpvA, FecA and HasR (references 30-32).

18) Ln 325: The recent complex structure of an STN with a sigma regulator should be referenced here. J Biol Chem. 2020 Apr 24;295(17):5795-5806. doi: 10.1074/jbc.RA119.010697.

Referenced as requested (reference 34).

19) Ln 331: There is no similarity of the FoxA STN to the NTE. This needs to be made more clear from a proper protein structure point of view.

We agree that the structures are different and have changed the text to reflect this (lines 357-359). With this change we don't think that it needs to be made more clear.

20) Ln. 343: "in several Sus" does not make sense. Please expand.

This has been clarified in the text (lines 368-371).

21) Ln 349: DP is undefined.

DP was defined as degree of polymerisation on line 64 of the original manuscript.

22) Ln 370: What is the evidence for beta 2,1 decorations? Is there another Levan that has more?

Levans have previously been shown by methylation analysis to contain beta 2,1 branches to varying degrees (~5-10% 2,1 vs 2,6 links - see e.g. Blake et al 1982; Benigar et al 2014 - these are now referenced in the revised MS, line 74) and we also show that a 2,1 decorated FOS is the most likely ligand bound to the CD complex (at both FOS1 and FOS 2 sites; Fig 1 & 2). The amount of 2,1 branching in levan from *Erwinia herbicola* has previously been shown to be ~10%, which is similar to the level seen in other bacterial levans such as from *Zymomonas* and *Bacillus* spp. (Benigar et al 2014). We have now added this info to the revised MS, lines 144-146.

Notably, the presence of a discrete 2,1 decoration in the FOS bound to the CD complex is likely biologically relevant as the FOS used is a product of Bt1760 endo-levanase digestion of *Erwinia* levan.

23) Ln 375: What is the concentration of the FOS in the media?

The molar concentration of FOS in the media when grown on levan is not known as discussed in point #16 above. In addition, in WT cells the surface endo-levanase will be actively cleaving longer FOS - thereby decreasing the average DP of the FOS and effectively increasing its molar concentration prior to import.

Is it possible that it is being pushed through on a concentration gradient, which would make the binding unnecessary? This possibility should probably be mentioned.

Passive diffusion of FOS is not possible with a native, plugged SusC. In other words, substrate binding to the transporter (Bt1763) is necessary to generate the interaction with TonB leading to channel formation, but binding to the SusD (Bt1762) may indeed be

dispensable if there is a lot of FOS outside the cell. We have added this possibility to the revised text (lines 406-409).

24) Ln 397: Why not perform the ITC at 6-fold concentration to see binding? This would remove ambiguity.

This is true and it is an experiment we would like to try. However, our ITC machine broke down about a year ago, and all of the reported ITC data were collected on a demo machine that we do not have anymore. The current lockdown and accessibility of lab facilities makes acquiring the data in another department very difficult. Our feeling is that we would see some weaker binding to T114 (compared to T115) at a higher ligand concentration, which would be consistent with the MS data.

25) Ln 411-422: Why not grow the cultures on media that has been depleted of smaller fragments by growth of the mutant? Essentially grow the mutant twice on the same media after removal of the bacteria.

The spent media from growth of the delta1760 mutant on MM-levan does not support further growth of the mutant strain. We have tried analysing the spent media from the mutant strain to determine the minimum size of the remaining FOS, but this was not possible by HPAEC-PAD (high performance anion exchange chromatography with pulsed amperometric detection) or MS due to the technical limitations of separating FOS at this high DP (added as lines 453-456).

26) Fig. 7. Is confusing. In 7a the +33 peak is labeled, but in 7b the +32 peak is labeled. Why did the authors switch from +33 to +32? Just label the +32 peak to avoid confusion.

Thank you for the suggestion. The figure labeling has now been changed.

27) Ln 490: It seems that a concentration gradient could also explain this.

See our response to point 23 above.

28) Ln 587: FUdR is undefined.

29) Ln 593: BHI is defined, do not spell out.

30) Ln 595: MM is already defined.

31) Ln 597: TSB is not defined.

32) Ln 606: What is the concentration of LDAO? It is not defined.

All these changes have now been made.

33) Ln 622: Buffer exchange is not detailed. This is one of the places where the presentation of the detail in the methods is incongruous. The authors should make sure that sufficient detail is provided to reproduce the experiments. That detail is lacking here.

The paragraph from line 644-647 has been edited to add detail. No buffer exchange step was used and we have now changed this.

34) Ln 714: Presumably the authors used U-15N and U-13C to obtain uniformly labeled protein for the NMR experiments.

Yes, but this was already mentioned in the original methods (line 727) and main text (line 296).

35) Ln 812: The authors have not reported the NMR data deposition. The data should be deposited at the BMRB - Biological Magnetic Resonance Bank and reported here with the other data deposition information. This needs to be done prior to acceptance.

This information has now been included.

36) Extended Fig 1: This figure should be remade. The cartoon representation is not useful when trying to determine the quality of electron density as it is designed to be a smoothed representation of the backbone rather than an accurate description. The authors should either use a true Alpha backbone representation or a full atom “sticks” representation to show how the quality of the electron density.

The figures 1b and 1c have been changed as requested. For clarity, we have opted to show Alpha backbone representations.

37) Supplementary Table 1: The $I/\sigma I$ for the outermost shell is quite low. It is presumed that the authors are using a CC1/2 cutoff to define the $I/\sigma I$ cutoff. This CC1/2 value should be reported for the outermost shell or the resolutions should be cut back to $I/\sigma I = 2.0$. If the CC1/2 was used then reporting this would show that the data with $I/\sigma I < 2$ does actually have useful information. Otherwise it is an erroneous assumption that data that is poorly measured contributes positively to the model. Hopefully the authors performed a paired refinement to get a true CC1/2. If they haven't, this is trivial to do in Phenix and should be done prior to acceptance.

The CC1/2 values have now been reported in the table. For all datasets, the CC1/2 values for the highest resolution shells (0.48, 0.63 and 0.60) are clearly acceptable (generally considered to be $CC1/2 > 0.3$). We therefore used all collected reflections for refinement (until the edge of the detector), and there was no need to determine resolution cutoffs via paired refinements.

REVIEWERS' COMMENTS

Reviewer #1 (Remarks to the Author):

The authors have satisfactorily addressed all of my comments and this is now a very strong paper.

Reviewer #2 (Remarks to the Author):

The authors answered all my comments satisfactorily. Congratulations on this very nice manuscript.

Reviewer #3 (Remarks to the Author):

The authors have sufficiently addressed the points raised in review, except for the referencing of the STN domains. REF 34 should be grouped with REF 30-32 as it is the relevant structure showing interaction between a STN domain and an anti-sigma factor.

This minor issue should be trivial to address. Nice work.